# OVERCOMING THE SPECTRAL BIAS OF NEURAL VALUE APPROXIMATION

**Ge Yang**[*†‡], **Anurag Ajay**[*‡§] & **Pulkit Agrawal**[†‡§]
NSF AI Institute for Artificial Intelligence and Fundamental Interactions (IAIFI)[†]
Computer Science and Artificial Intelligence Laboratory (CSAIL) [‡]
Improbable AI Lab[§]
Massachusetts Institute Technology

## ABSTRACT

Value approximation using deep neural networks is at the heart of off-policy deep reinforcement learning, and is often the primary module that provides learning signals to the rest of the algorithm. While multi-layer perceptron networks are universal function approximators, recent works in neural kernel regression suggest the presence of a *spectral bias*, where fitting high-frequency components of the value function requires exponentially more gradient update steps than the low-frequency ones. In this work, we re-examine off-policy reinforcement learning through the lens of kernel regression and propose to overcome such bias via a composite neural tangent kernel. With just a single line-change, our approach, the Fourier feature networks (FFN) produce state-of-the-art performance on challenging continuous control domains with only a fraction of the compute. Faster convergence and better off-policy stability also make it possible to remove the target network without suffering catastrophic divergences, which further reduces TD$(0)$'s estimation bias on a few tasks. Code and analysis available at https://geyang.github.io/ffn.

## 1 INTRODUCTION

At the heart of reinforcement learning is the question of how to attribute credits or blame to specific actions that the agent took in the past. This is referred to as the *credit assignment* problem (Minsky, 1961). Correctly assigning credits requires reasoning over temporally-extended sequences of observation and actions, so that trade-offs can be made, for example, to choose a less desirable action at a current step in exchange for higher reward in the future. Temporal difference methods such as TD$(\lambda)$ and Watkins' Q-learning (Sutton, 1988; Watkins, 1989) stitch together the immediate rewards local to each state transition, to estimates the discounted sum of rewards over longer horizons. This is an incremental method for dynamic programming (Watkins & Dayan, 1992) that successively improves the value estimate at each step, which reduces the computation that would otherwise be needed to plan ahead at decision time.

A key tension in scaling TD learning to more challenging domains is that the state space (and action space in continuous control problems) can be very large. In Neurogammon (Tesauro, 1991), for example, the complexity of the state space is on the scale of $\mathcal{O}(10^{20})$, which prevents one from storing all state-action pairs in a table. Such constraints and the need to generalize to unseen board arrangements prompted Tesauro (1991) to replace the look-up table with an artificial neural network to great effect, achieving master-level play on backgammon. In general, however, adopting neural networks as the value approximator introduces learning instability that can sometimes lead the iterative learning procedure into a divergent regime where errors are amplified at each step (Sutton & Barto, 2018; Bertsekas, 1995; Baird, 1995). A slew of algorithmic fixes have been developed to tackle this problem from different angles, including sampling from a replay buffer (Lin, 1992; Riedmiller, 2005a); using a delayed copy for the bootstrapping target (Mnih et al., 2013); and using ensembles (Hasselt, 2010; Lee et al., 2021).

Despite of the popularity of these techniques, the neural value approximator itself remains unchanged. Popular treatments view these networks as *universal* function approximators that would

---

[*] Equal contribution, order determined by rolling a dice. Correspondence to `{geyang,aajay}@csail.mit.edu`

*eventually* converge to arbitrary target functions given sufficient number of gradient updates and training data. In practice, however, state-of-the-art off-policy algorithms interleave optimization with sampling, which greatly limits the amount of optimization that is accessible. This puts deep reinforcement learning into the same regime as a neural network regularized through *early-stopping* where the model bias, the amount of optimization available and the sample size interact with each other in intricate ways (Belkin et al., 2018; Nakkiran et al., 2019; Canatar et al., 2021). Therefore to understand off-policy learning with neural value approximators we first need to understand how deep neural networks generalize, and how they converge under the dynamic process of gradient decent. Then to find a solution, we need to figure out a way to control the generalization and learning bias, so that deliberate bias-variance trade-offs could be made for better convergence.

Recent efforts in deep learning theory and computer graphics offer new tools for both. Jacot et al. (2018) uncovers a *spectral-bias* for shallow, infinitely-wide neural networks to favor low-frequency functions during gradient descent (Kawaguchi & Huang, 2019; Bietti & Mairal, 2019; Ronen et al., 2019). This result has since been extended to deeper networks at finite-width of any "reasonable" architecture via the *tensor program* (Yang, 2019; 2020; Yang & Littwin, 2021; Yang & Salman, 2019). Our analysis on the toy MDP domain (Section 3) shows that value approximator indeed underfits the optimal value function, which tends to be complex due to the recursive nature of unrolling the dynamics. Learning from recent successes in the graphics community (Sitzmann et al., 2020; Tancik et al., 2020), we go on to show that we can overcome the spectral-bias by constructing a composite kernel that first *lifts* the input into random Fourier features (Rahimi & Recht, 2008; Yang et al., 2015). The resulting neural tangent kernel is tunable, hence offering much needed control over how the network interpolates data.

Our main contributions are twofold. First, we show that the (re-)introduction of the Fourier features (Konidaris et al., 2011) enables faster convergence on higher frequency components during value approximation, thereby improving the sample efficiency of off-policy reinforcement learning by reducing the amount of computation needed to reach the same performance. Second, our improved neural tangent kernel produce *localized* generalization akin to a Gaussian kernel. This reduces the cross-talk during gradient updates and makes the learning procedure more stable. The combined effect of these two improvements further allows us to remove the target network on a few domains, which leads to additional reduction in the value estimation bias with TD(0).

## 2 BACKGROUND AND NOTATION

We consider an agent learning to act in a Markov Decision Process (MDP), $\langle S, A, R, P, \mu, \gamma \rangle$ where $S$ and $A$ are the state and action spaces, $P : S \times A \mapsto S$ is the transition function, $R : S \times A \mapsto \mathbb{R}$ is the reward and $\mu(s)$ is the initial state distribution. We consider an infinite horizon problem with the discount factor $\gamma$. The goal of the agent is to maximize the expected future discounted return $\mathcal{J} = \mathbb{E}\left[\sum_{t=0}^{\infty} \gamma^t R(s_t, a_t)\right]$ by learning a policy $\pi(a|s)$ that maps a state $s$ to a distribution over actions. The state-action value function (Q-function) is defined as $Q^\pi(s, a) = \mathbb{E}\left[\sum_{t=0}^{\infty} \gamma^t R(s_t, a_t)|(s_0, a_0) = (s, a)\right]$. The optimal $Q^*(s, a)$ is the fixed point of the Bellman optimality operator $\mathcal{B}^*$

$$\mathcal{B}^* Q(s, a) = R(s, a) + \gamma \mathbb{E}_{s' \sim P(s'|s,a)}[\max_{a*} Q(s', a^*)]. \quad (1)$$

The fitted Q iteration family of algorithms (Ernst et al., 2003; Riedmiller, 2005a) iteratively finds the (sub) optimal $Q_{\theta*}$ by recursively applying the gradient update

$$\theta' = \theta - \eta \nabla_\theta \mathbb{E}_{(s,a,s') \sim \mathcal{D}} \left\| Q_\theta(s, a) - \mathcal{B}^* Q_\theta(s, a) \right\|_2^2. \quad (2)$$

Let $\mathcal{X} = Q_\theta(s, a) - \mathcal{B}^* Q_\theta(s, a)$ and apply the chain-rule. The update in Equation 2 becomes

$$\theta' = \theta - 2\eta \mathbb{E}_{(s,a,s') \sim \mathcal{D}} \left[ \mathcal{X} \nabla_\theta Q_\theta(s, a) \right] \quad (3)$$

We can approximate the updated $Q$ in function form through its first-order Taylor expansion

$$Q_{\theta'}(s, a) \approx Q_\theta(s, a) - 2\eta \mathbb{E}_{(s',a') \sim \mathcal{D}} \left[ \mathcal{K}(s, a; s', a') \mathcal{X} \right] \quad (4)$$

where the bilinear form $\mathcal{K}(s, a; s', a') = \nabla_\theta Q_\theta(s, a)^T \nabla_\theta Q_\theta(s', a')$ is the *neural tangent kernel* (NTK, see Jacot et al. 2018 and Section 4). This procedure can be interpreted as producing a

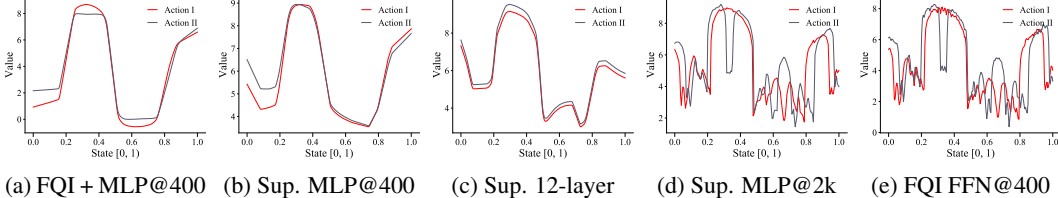

(a) FQI + MLP@400    (b) Sup. MLP@400    (c) Sup. 12-layer    (d) Sup. MLP@2k    (e) FQI FFN@400

Figure 2: Q-value approximation on the toy MDP. All baselines collected at 400 epochs unless otherwise mentioned. (a) Fitted Q iteration using a 4-layer MLP with 400 hidden neurons. (b) Supervised by the ground-truth value target(c) a larger network that is $3\times$deeper (12-layers). (d) the same 4-layer network, but optimized longer, for 2000 epochs. (e) 4-layer FFN, no target network.

sequence $\{Q_0, Q_1, Q_2, \dots\}$ using the iterative rule, $Q_{i+1} = \Pi\mathcal{B}^*(Q_i)$ starting with $Q_0(s,a) = R(s,a)$. $\Pi$ represents a projection to functions expressible by the class of function approximators, which reduces to the *identity* in the tabular case. The $i$th item in the sequence, $Q_i$, is the projected optimal $Q$ function of a derived MDP with a shorter horizon $H = i$.

# 3   A MOTIVATING EXAMPLE

We motivate through a toy MDP adapted from Dong et al. (2020), and show that neural fitted Q iteration significantly underfits the optimal value function. Despite of the simple-looking reward and forward dynamics, the value function is quite complex. Our spectral analysis in Section 3.1 indicates that such complexity arises from the recursive application of the Bellman operator.

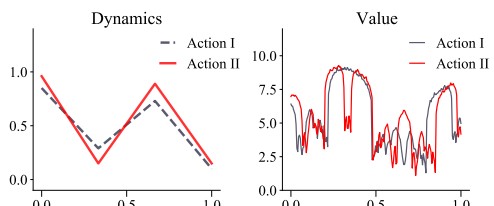

Figure 1: A Toy MDP with a simple forward dynamics and complex value function, adapted from Dong et al. (2020).

**Toy MDP Distribution**   Consider a class of toy Markov decision processes $M$. The state space is defined on the real-line as $\mathcal{S} = \mathbb{R}^{[0,1)}$. The reward is the identity function. The action space is a discrete set with two elements $\{0, 1\}$, each corresponds to a distinct forward dynamics that is randomly sampled from the space of piece-wise linear functions with $k$ "kinks." For all of the examples below, we use a fixed number $k = 10$, and uniformly sample the value of each turning point in this dynamic function between 0 and 1. The result is a distribution $\mathcal{M}$ that we can sample from

$$M \sim \mathcal{M} = p(M). \tag{5}$$

**A Curious Failure of Neural Fitted Q Iteration**   When we apply standard neural fitted Q iteration (FQI, see Riedmiller 2005b) to this toy problem using a simple four-layer multi-layer perceptron (MLP) with 400 hidden units at each layer, we observe that the learned MLP value approximator, produced in Figure 2a, significantly underfits in comparison to the optimal value function. One hypothesis is that Q-learning is to blame, and in fact prior works such as Kumar et al. 2021 have argued that underfitting comes from minimizing the TD (Temporal-Difference) error, because bootstrapping resembles self-distillation (Mobahi et al., 2020), which is known to lead to under-parameterization in the learned neural network features. To show that Q learning is not to blame, we can use the same MLP architecture, but this time directly regress towards the (ground truth) optimal Q function via supervised learning. Figure 2b shows that the under-fitting still persists under supervised learning, and increasing the depth to 12 layers (see Figure 2c) fails to fix the problem. This shows an over-parameterized network alone is insufficient to reduce under-fitting. Finally, if we increase the number of training iterations from 400 epochs to 2000, the original four-layer MLP attains a good fit. This shows that solely focusing on the expressivity of the neural network, and making it bigger (Sinha et al., 2020) can be misguided. The class of function that can be approximated depends on how many gradient updates one is allowed to make, a number capped by the compute budget.

## 3.1   SPECTRAL SIGNATURE OF THE TOY MDP

The Bellman optimality operator imprints a non-trivial spectrum on the resulting $Q_i$ as it is applied

recursively during fitted Q iteration. We present the evolving spectra marginalized over $\mathcal{M}$[1] in Figure 3. As the effective horizon increases, the value function accrues more mass in the higher-frequency part of the spectrum, which corresponds to less correlation between the values at near-by states. In a second experiment, we fix the horizon to 200 while increasing the discount factor from 0.1 all the way up to 0.99. We observe a similar whitening of the spectrum at longer effective recursion depths. In other words, the complexity of the value function comes from the repeated application of the Bellman optimality operator in a process that is not dissimilar to an "infinity mirror." The spectrum of the Bellman operator gets folded into the resulting Q function upon each iterative step. Although our analysis focuses on the state space, the same effect can be intuitively extrapolated to the joint state-action space.

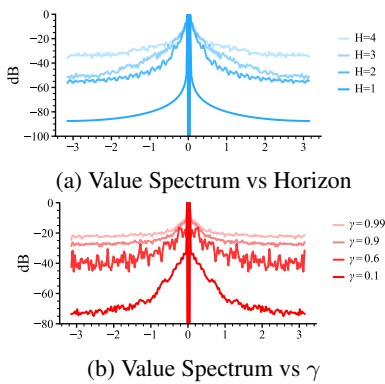

(a) Value Spectrum vs Horizon

(b) Value Spectrum vs $\gamma$

Figure 3: The spectrum of the optimal value function noticeably whitens over (a) longer horizon and (b) larger $\gamma$.

## 3.2 KERNEL VIEW ON OFF-POLICY DIVERGENCE

Following the formulation of convergence in Dayan (1992); Tsitsiklis (1994); Jaakkola et al. (1994):

**Definition:** Consider a complete metric space $S$ with the norm $\|\cdot\|$. An automorphism $f$ on $S$ is a *contraction* if $\forall a, b \sim S$, $\|f(a) - f(b)\| \leq \gamma\|a - b\|$. Here $\gamma \in [0,1)$ is called the *contraction modulus*. When $\gamma = 1$, $f$ is a *nonexpansion*.

**Banach fixed-point theorem.** Let $S$ be non-empty with a contraction mapping $f$. Then $f$ admits a unique fixed-point $x^* \in S$ *s.t.* $f(x^*) = x^*$. Furthermore, $\forall x_0 \in S$, $x^*$ is the limit of the sequence given by $x_{i+1} = f(x_i)$. *a.k.a* $x^* = \lim_{i \to \infty} x_i$.

Without lost of generality, we can discretize the state and action space $S$ and $A$. The NTK becomes the gram matrix $\mathcal{K} \in \mathbb{R}^{|S \times A| \times |S \times A|}$. Transition data are sampled from a distribution $\rho(s, a)$.

**Theorem.** *(Achiam et al., 2019) Let indices $i$, $j$ refer to state-action pairs. Suppose that $\mathcal{K}$, $\eta$ and $\rho$ satisfy the conditions:*

$$\forall i, 2\eta\mathcal{K}_{ii}\rho_i < 1, \tag{6}$$

$$\forall i, (1 + \gamma) \sum_{j \neq i} |\mathcal{K}_{ij}|\rho_j \leq (1 - \gamma)\mathcal{K}_{ii}\rho_i, \tag{7}$$

*Then, Equation 4 induces a contraction on Q in the sup norm, with fixed-point Q\* and the TD loss optimization converges with enough optimization steps.*

For relatively large $\gamma$ (for instance, $\gamma \in (0.99, 0.999)$), the above theorem implies that small off-diagonal terms in the NTK matrix are sufficient conditions for convergence.

## 4 SPECTRAL-BIAS AND NEURAL KERNEL REGRESSION

Consider a simple regression problem where we want to learn a function $f(\xi; \theta) \in \mathbb{R}$. $\xi$ is a sample from the dataset. To understand how the output of the network changes w.r.t small perturbations to the parameters, we can Taylor expand around $\theta$

$$f(\xi; \theta + \delta\theta) - f(\xi; \theta) \approx \langle \nabla\theta f(\xi; \theta), \ \delta\theta \rangle. \tag{8}$$

During stochastic gradient descent using a training sample $\hat{\xi}$ with a loss function $\mathcal{L}$, the parameter update is given by the product between the loss derivative $\mathcal{L}' \circ f(\hat{\xi})$ and the *neural tangent kernel* $\mathcal{K}$ (Jacot et al., 2018)

$$\delta\theta = -\eta\mathcal{L}'(f(\hat{\xi}))\mathcal{K}(\xi, \hat{\xi}) \quad \text{where} \quad \mathcal{K}(\xi, \hat{\xi}) = \langle \nabla_\theta f(\xi; \theta), \nabla_\theta f(\hat{\xi}; \theta) \rangle. \tag{9}$$

---

[1]In this distribution, we modify the reward function to be $\sin(2\pi s)$ to simplify the reward spectrum.

In the infinite-limit with over-parameterized networks, the function remains close to initialization during training (Chizat et al., 2019). The learning dynamics in this "lazy" regime, under an $L_2$ regression loss behaves as a minimum norm least-square solution

$$f_t - f^* = e^{-\eta \mathcal{K} t(\xi, \hat{\xi})}(f_0 - f^*),$$ (10)

where $f_t$ is the function under training at time $t$ and $f_0$ is the neural network at initialization. Replacing $\mathcal{K}$ with the Gramian matrix $\mathcal{K}$ between all pairs of the training data, we can re-write Equation 4 in its spectral form $\mathcal{K} = \boldsymbol{O}\boldsymbol{\Lambda}\boldsymbol{O}^T$ where each entry in the diagonal matrix $\boldsymbol{\Lambda}$ is the eigenvalue $\lambda_i > 0$ for the basis function $O_i$ in the orthogonal matrix $\boldsymbol{O}$. Using the identity $e^{\boldsymbol{A}} = \boldsymbol{O}e^{\boldsymbol{\Lambda}}\boldsymbol{O}^T$, we can decompose the learning dynamics in Equation 4 into

$$\boldsymbol{O}^T(f_t - f^*) = e^{-\eta \boldsymbol{\Lambda} t}\boldsymbol{O}^T(f_0 - f^*).$$ (11)

The key observation is that the convergence rate on the component $O_i$, $\eta\lambda_i$, depends exponentially on its eigenvalue $\lambda_i$. *a.k.a* the absolute error

$$|O_i^T(f_t - f^*)| = e^{-\eta\lambda_i t}|O_i^T(f_0 - f^*)|$$ (12)

Multiple works (Rahaman et al., 2019; Shah et al., 2020; Yang & Salman, 2019; Huh et al., 2021) have shown that the NTK spectrum ($\lambda_i$) of a regular ReLU network decays rapidly at higher frequency. In particular, Bietti & Mairal (2019) provided a bound of $\Omega(k^{-d-1})$ for the $k$th spherical harmonics[2]. Such results have been extended to finite-width networks of arbitrary architecture and depth via the *tensor programs* (Yang, 2019; 2020).

The Gramian matrix form of the NTK also offers an intuitive way to inspect state-aliasing during gradient descent. This is because the off-diagonal entries corresponds to the similarity between gradient vectors for different state-action pairs. The kernel of an multi-layer perceptron is not stationary when the input is not restricted to a hypersphere as in Lee et al. (2017). We can, however, compute $\mathcal{K}$ of popular network architectures over $\mathbb{R}^{[0,1)}$, shown in Figure 4. The spectral-bias of the NTK of both ReLU networks and hyperbolic tangent networks produce large off-diagonal elements, increasing the chance of divergence.

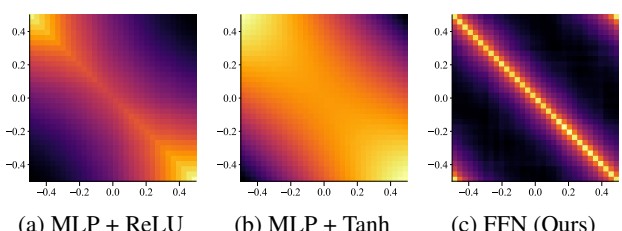

(a) MLP + ReLU    (b) MLP + Tanh    (c) FFN (Ours)

Figure 4: NTK comparison between (a) MLP with ReLU activation, (b) MLP with tanh activation, and (c) Fourier feature networks (Ours). The MLP NTK in (a,b) both contain large off-diagonal elements. The addressing by the gradient vectors is not specific to each datapoint.

## 5 OVERCOMING THE SPECTRAL-BIAS OF NEURAL VALUE APPROXIMATION

To correct the spectral-bias of a feed-forward neural network, we construct a composite kernel where a random map $\boldsymbol{z} : \mathbb{R}^d \mapsto \mathbb{R}^D$ first "*lifts*" the input into a randomized harmonics basis. This explicit kernel lifting trick was introduced by Rahimi & Recht (2007) and it allowed the authors to fit complex datasets using a linear machine. The mixing brings high-frequency input signals down to a lower and more acceptable band for the ReLU network. Data also appears more sparse in the higher-dimensional spectral basis, further simplifies learning. To emulate arbitrary shift-invariant kernel $\mathcal{K}^*$, Rahimi & Recht (2007) offered a procedure that sample directly from the nominal distribution given by $\mathcal{K}^*$'s spectrum $\mathscr{F}(\mathcal{K}^*)$

$$k(x,y) = \langle \phi(\xi), \phi(\hat{\xi}) \rangle \approx \boldsymbol{z}(\xi)^T \boldsymbol{z}(\hat{\xi}) \text{ where } z(\xi) = \sum_i w_i e^{2\pi k_i} \text{ and } w_i \sim \mathscr{F}(\mathcal{K}^*).$$ (13)

We choose to initialize FFN by sampling the weights from an isotropic multivariate Gaussian with a tunable cutoff frequency $b$. We modify the normalization scheme from Rahimi & Recht (2007) and divide $b$ with the input dimension $d$, *s.t.* similar bandwidth values could be applied across a wide

---

[2] the input is restricted to a sphere. This bound is loose, and for a tighter bound refer to Cao et al. (2019).

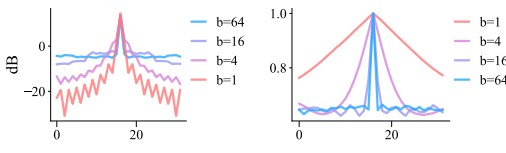

(a) FFN kernel Spectrum  (b) NTK Cross-section

Figure 5: FFN provide direct control over generalization. (a) The kernel spectra. Peak in the center is due to windowing effect. Higher band-limit leads to flatter spectra. (b) The cross-section of the kernel at different band-limit.

**Algorithm** Learned Fourier Features (LFF)

```
class LFF(nn.Linear):
  def __init__(self, in, out, b_scale):
    super().__init__(in, out)
    init.normal_(self.weight, std=b_scale/in)
    init.uniform_(self.bias, -1.0, 1.0)

  def forward(self, x):
    x = np.pi * super().forward(x)
    return torch.sin(x)
```

variety of reinforcement learning problems with drastically different state and action space dimensions. We slightly abuse the notion by using $b_j$ as the bias parameters, and $b$ (without subscript) for the *bandwidth*

$$\text{RFF}(x)_j = \sin(\sum_{i=1}^{d} w_{i,j}x^i + b_j) \text{ where } w_{i,j} \sim \mathcal{N}(0, \pi b/d) \text{ and } b_j \sim \mathcal{U}(-\pi, \pi). \quad (14)$$

To attain better performance, Rahimi & Recht (2008) learns a *weighted sum* of these random kitchen sink features, whereas Yang et al. (2015) adapts the sampled spectral mixture itself through gradient descent. Using the modern deep learning tool chain, we can view the entire network including the random Fourier features as a *learnable* kernel. For small neural networks with limited expressivity, we found that enabling gradient updates on the RFF parameters is important for performance. We refer to this adaptive variant as the *learned Fourier features* (LFF), and the shallow MLP with an adaptive Fourier feature expansion as the **Fourier feature networks** (FFN).

**Kernel Spectrum, Convergence, and Interpolation** The composite kernel of the FFN has larger eigenvalues on the high-frequency harmonics (see Figure 5), which improves the speed at which the network converges. We can tune this kernel spectrum by changing the band-limit parameter $b$. With a larger bandwidth, the interpolation becomes more *local*. On the toy domain used to motivate this work FFN achieves a perfect fit with just 400 optimization epochs whereas the MLP baseline requires at least two thousand gradient steps (see Figure 2e). The gain is even more prominent on more challenging environments in the results section (Quadruped run, see Figure 9). Optimization overhead is a key bottleneck in off-policy learning, therefore faster convergence also translates into better sample efficiency, and faster wall-clock time.

**Improving Off-policy Stability** "cross-talks" between the gradients of near-by states is a key factor in off-policy divergence. Such "cross-talk" manifest as similarity between gradient vectors, the measure of which is captured by the Gram matrix of the NTK. The Fourier feature network kernel is both *localized* (Figure 4c) and *tunable* (Figure 5), offering direct control over the bias-variance trade-off. The improved learning stability allows us to remove the target network on a number of domains while retaining substantial performance. A curious find is when

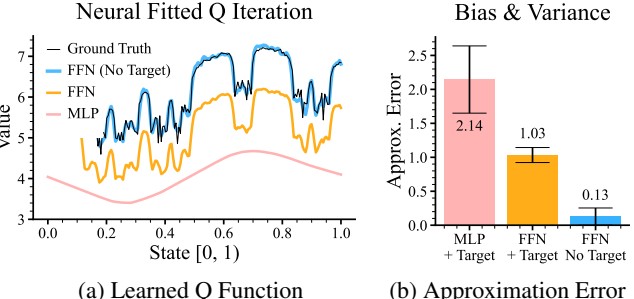

(a) Learned Q Function  (b) Approximation Error

Figure 6: (a) Comparison of the learned Q function (for action II). (b) Approximation error averaged over 10 random seeds. FFN with a target network still contains an offset, whereas removing the target network eliminates such bias.

the target network is used, the learned value approximation still contains a large constant offset from the optimal value function (yellow curve in Figure 6). Such offset disappears after we remove the target value network (blue curve).

**Visualizing Mountain Car** The Mountain Car environment (Moore, 1990; Sutton, 2000) has a simple, two-dimensional state space that can be directly visualized to show the learned value func-

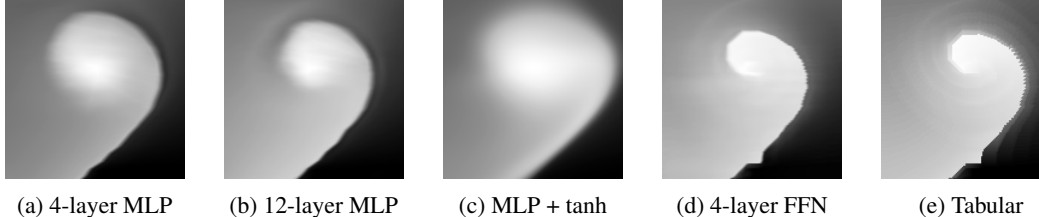

(a) 4-layer MLP     (b) 12-layer MLP     (c) MLP + tanh     (d) 4-layer FFN     (e) Tabular

Figure 7: Visualizing the learned value function for action $0$ in Mountain Car using (a) a four-layer ReLU network, (b) an eight-layer network, (c) a four-layer network with hyperbolic-tangent activation, which is commonly used in deep reinforcement learning. (d) that of a four-layer FFN. (e) shows the ground-truth optimal value function produced by running tabular value iteration. The two axis corresponds to velocity (horizontal) and position (vertical) of the car.

tion. We compare the value estimate from three baselines In Figure 7: the ground-truth value estimate acquired using tabular value iteration; one obtained from a four-layer ReLU network using fitted Q iteration; and one from the same network, but using a 16 dimensional Fourier features on the input. All networks use 400 latent neurons and are optimized for 2000 epochs.

## 6   SCALING UP TO COMPLEX CONTROL TASKS

We scale the use of FFN to high-dimensional continuous control tasks from the DeepMind control suite (Tassa et al., 2020) using soft actor-critic (SAC, Haarnoja et al. 2018) as the base algorithm. Our implementation is built off the pytorch codebase from Yarats & Kostrikov (2020) which is the current *state-of-the-art* on state space DMC. Both the actor and the critic have three latent layers in this base implementation. We use FFN for both the actor and the critic by replacing the first layer in the MLP with learned Fourier features (LFF) without increasing the total number of parameters.

We provide extensive empirical analysis on eight common DMC domains and additional results with DDPG in Appendix A.9. Due to space constraints we present the result on four representative domains in Figure 8. FFN consistently improves over the MLP baseline in complex domains while matching the performance in simpler domains. We include additional ablations and hyperparameter sensitivity analysis in Appendix A.7.

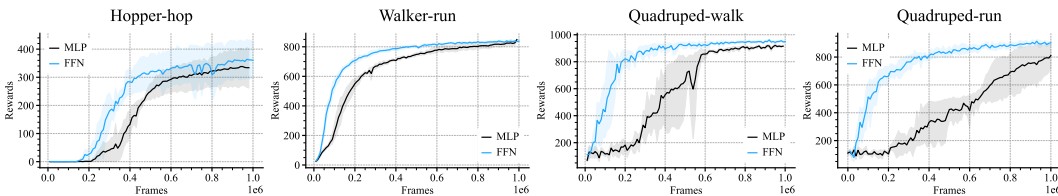

Figure 8: Learning curve with FFN applied to *SAC* on the DeepMind control suite. Domains are ordered by input dimension, showing an overall trend where domains with higher state dimension benefits more from Fourier features. We use a (Fourier) feature-input ratio of $40 : 1$.

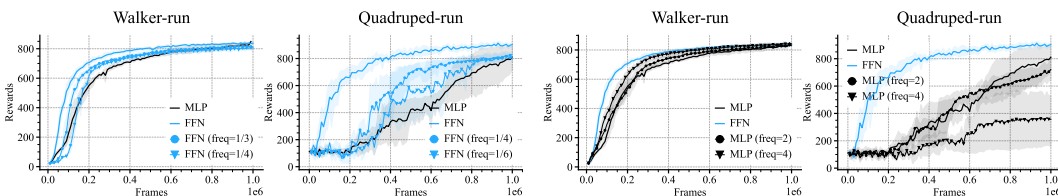

Figure 9: FFN needs only a fraction of the compute to match the performance of the MLP baseline on both Walker-run and Quadruped-run.

Figure 10: Increasing the gradient updates to sampling ratio causes Quadruped-run to crash. On Walker-run it improves the performance.

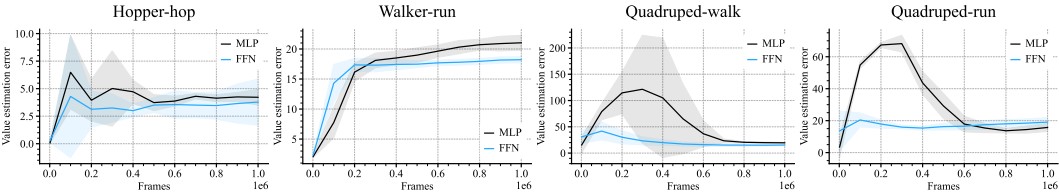

Figure 11: *Weight and bias changes* in FFN and MLP during training, using *SAC*. While FFN's bias parameters undergo less change than MLP's bias parameters, the results are mixed when it comes to weight parameters on the Quadruped environment.

**Faster Convergence via Fourier Feature Networks**   The key benefit of the Fourier feature network is that it reduces the computation needed for good approximation. Off-policy reinforcement learning algorithms tend to be bottlenecked by optimization whereas on-policy algorithms tend to be bottlenecked by simulation time. This means with a reduced replay ratio, FFN can achieve faster wall-clock time. Figure 9 shows that we can indeed reduce the update frequency to $1/4$ in *Walker-walk* and $1/6$ in *Quadruped-run* while still matching the performance of the MLP baseline. In a control experiment (see Figure 10), we find that increasing the replay ratio causes the MLP to crash on Quadruped-run, illustrating the intricate balance one otherwise needs to maintain. FFN brings additional improvements on learning stability besides faster convergence. We additionally visualize the value approximation error in Figure 25. FFN consistently produces smaller value approximation error compare to the MLP.

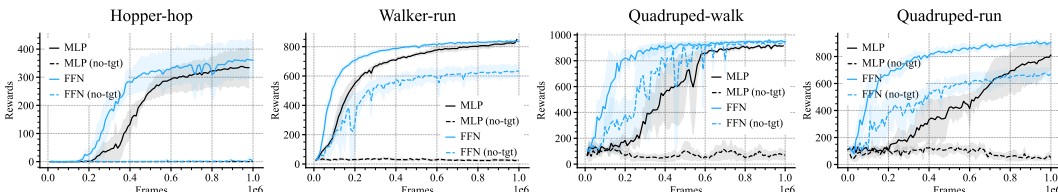

Figure 12: Value approximation error with FFN vs MLP using *SAC*. The divergence is especially prominent at the beginning of learning when the data is sparse. FFN reduces the variance in these regimes by making more appropriate bias-variance trade-off than the MLP baseline.

**Improved Feature Representation Makes Learning Easier**   The random Fourier features sample from a Gaussian distribution in the frequency domain. This is a more uniform functional distribution compared to the MLP, which exponentially favors low-frequency functions (Bietti & Mairal, 2019). Let $\mathcal{W}_t$ and $\mathcal{B}_t$ refer to the concatenated weight and bias vectors. In this experiment we inspect the $\ell^2$-norm of the changes of weights and biases *w.r.t* to their initial value, $\|\mathcal{W}_t - \mathcal{W}_0\|_2$ and $\|\mathcal{B}_t - \mathcal{B}_0\|_2$. We include all eight DMC domains in Appendix A.11 and present the most representative domains in Figure 11. In seven out of eight domains, the first layer in both the MLP and the FFN experience roughly the same amount of change in the weights, but the FFN requires less weight change in the later layers. This indicates that LFF makes learning easier for the rest of the network. Quadruped-run is the only outlier where the LFF experience more weight change than the first layer of the MLP. Our hypothesis is that this is a harder task, and we are either underparameterized in terms of Feature dimension, or there is a distribution-misalignment between the task and the model. So far we have focused on the weights. The biases change consistently less in the FFN than the MLP.

Figure 13: Learning curves showing that FFN improves learning stability to the extent that learning can happen on some domains even without the target network. On the contrary, MLP consistently fails without a target value network. FFN has consistently less variance than the MLP.

Figure 14: Control from pixels learning curve on the DeepMind control suite using Fourier-CNN (F-CNN) and *DrQv2*. We use a feature-input ratio of $40 : 1$. Performance with the F-CNN has much lower variance and is consistently at the top of the confidence range of the vanilla CNN.

**Fourier Features Improve Off-Policy Stability** Without a target network, gradient updates through the TD objective causes the regression target itself to change, making the learning procedure less stable. Mnih et al. (2013) replaces the value network on the *r.h.s* of the Bellman equation with a copy that is updated at a slower rate, so that the TD objective is more stationary. In Figure 13 we remove the target network. While the MLP baseline completely fails in all environments, the improved neural tangent kernel of the FFN sufficiently stabilizes Q learning, that only minor performance losses occur with seven out of eight domains. We offer the complete result in Appendix A.12.

**Improving the Convolution Kernel** The filters in a standard convolution neural network suffer the same spectral bias in the RGB and feature dimension as the state and action space above. We can in principle extend our spectral-bias fix to convolution networks by replacing the first layer with a $1 \times 1$ convolution with a sine non-linearity. In this experiment we build upon the DrQv2 implementation (Yarats et al., 2021) and replace the CNN in both the actor and the critic with this Fourier-CNN (F-CNN) architecture according to Equation 14. The results indicate that while F-CNN's performance is within the variance of CNN's performance, its variance is much lower (see Figure 14). Such technique has also been used by Kingma et al. (2021) to improve image generation with diffusion models. For more detailed theoretical analysis of the CNN NTK, one can refer to Arora et al. (2019) and Li et al. (2019).

## 7 DISCUSSION

The inspiration of this work comes from our realization that techniques used by the graphics community (Tancik et al., 2020; Sitzmann et al., 2020; Mildenhall et al., 2020) to learn high-fidelity continuous neural representations represent a new way to explicitly control generalization in neural networks. We refer to Achiam et al. (2019)'s pioneering work for the theoretical setup on off-policy divergence, which predates many of the recent analysis on spectral-bias in neural networks, and the (re-)introduction of the random Fourier features as a solution to correct such learning biases. Functional regularization through gradient conditioning is an alternative to re-parameterizing the network. A recent effort could be found in Piché et al. (2021). A key benefit of reparameterizing the network is speed. We managed to reduce the wall-time by $32\%$ on the challenging Quadruped environment, by reducing the replay ratio to $1/6$ of the baseline.

Fourier features for reinforcement learning dates back as early as Konidaris et al. (2011). While this manuscript was under review, a few similar publication and preprints came out, all developed independently. Li & Pathak (2021) focuses on the smoothing effect that Fourier feature networks have in rejecting noise. Our finding is that vanilla feed-forward neural networks are on the biased-end of the bias-variance trade-off. In other words, we believe the main issue with neural value approximation is that the network *underfits* real signal, as opposed to being overfit to random noise. The rank of the neural representation describes the portion of the linear space occupied by the largest eigenvalues of the kernel regardless of the spatial frequency of those corresponding eigenfunctions. Therefore lower rank does not correspond to smoother functions. We additionally find that maintaining a Fourier feature to input feature ratio ($D/d > 40$) is critical to the expressiveness of the network, which allowed us to scale up to Quadruped without needing to concatenating the raw input as a crutch. Brellmann et al. (2022) is a concurrent submission to ours that delightfully includes the random tile encoding scheme from Rahimi & Recht (2007), and on-policy algorithm, PPO. Our intuition is that policy gradient needs to be considered a life-long learning problem, and *locality* in the policy network can speed up learning by eliminating the need to repeatedly sample areas the policy has experienced. We are excited about these efforts from the community, and urge the reader to visit them for diverse treatments and broader perspectives.

## ACKNOWLEDGMENTS

The authors would like to thank Leslie Kaelbling for her feedback on the manuscript; Jim Halverson and Dan Roberts at IAIFI for fruitful discussions over neural tangent kernels, and its connection to off-policy divergence; Aviral Kumar at UC Berkeley for bringing into our attention the expressivity issues with off-policy value approximation; and the reviewer and area chair for their kind feedback.

This work is supported in part by the National Science Foundation Institute for Artificial Intelligence and Fundamental Interactions (IAIFI, https://iaifi.org/) under the Cooperative Agreement PHY-2019786; the Army Research Office under Grant W911NF-21-1-0328; and the MIT-IBM Watson AI Lab and Research Collaboration Agreement No.W1771646. The authors also acknowledge the MIT SuperCloud and Lincoln Laboratory Supercomputing Center for providing high performance computing resources. The views and conclusions contained in this document are those of the authors and should not be interpreted as representing the official policies, either expressed or implied, of the Army Research Office or the U.S. Government. The U.S. Government is authorized to reproduce and distribute reprints for Government purposes notwithstanding any copyright notation herein.

## REPRODUCIBILITY STATEMENT

We include detailed implementation details in Appendix A.4.

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

# A   APPENDIX

## A.1   EXPERIMENTAL DETAILS ON THE TOY MDP

**Offline Data Generation**   We divided the 1-dimensional state space into 1000 discrete bins and used the midpoints of the bins as initial states. We then took both the actions from these initial states to get corresponding next states. We used the dataset of these transitions (2000 total transitions) as our offline dataset.

**Optimization details**   We use 4-layer MLP with ReLU Activation, with 400 latent neurons. We use Adam optimization with a learning rate of 1e-4, and optimize for 400 epochs. We use gradient descent with a batch size of 200. For a deeper network, we use a 12-layer MLP, with 400 latent neurons but keep other optimization hyperparameters fixed. To show that longer training period help MLPs evade the spectral bias, we use a 4-layer MLP, with 400 latent neurons and train it for 2000 epochs. All the optimization hyperparameters are kept the same.

## A.2   EXPERIMENT DETAILS ON MOUNTAIN CAR

**Environment**   We use the implementation from the OpenAI gym (Brockman et al., 2016), and discretize the state space into 150 bins. This is a relatively small number because otherwise the space complexity becomes infeasible large. We use a four-layer MLP as the Q function, and a bandwidth parameter of 10.

## A.3   EXPERIMENTAL SETUP ON DEEPMIND CONTROL SUITE

**DMC domains**   We list the 8 DMC domains along with their observation space dimension and action space dimension in Table 1. We also list the optimal bandwidth $b$ used by FFN for each of these envs.

Table 1: DMC domains in increasing order of state space and action space dimensionality

| Name | Observation space | Action space | Bandwidth $b$ |
|------|-------------------|--------------|---------------|
| Acrobot-swingup | Box(6,) | Box(1,) | 0.003 |
| Finger-turn-hard | Box(12,) | Box(2,) | 0.001 |
| Hopper-hop | Box(15,) | Box(4,) | 0.003 |
| Cheetah-run | Box(17,) | Box(6,) | 0.001 |
| Walker-run | Box(24,) | Box(6,) | 0.001 |
| Humanoid-run | Box(67,) | Box(21,) | 0.001 |
| Quadruped-walk | Box(78,) | Box(12,) | 0.0003 |
| Quadruped-run | Box(78,) | Box(12,) | 0.0001 |

## A.4   ARCHITECTURAL DETAILS

For the state space problems, we parameterize the actor and the critic in the MLP baseline with three latent layers. For FFN (ours), we replace the first layer in the MLP a learned Fourier features layer (LFF). If $d$ is the input dimension, both MLP and FFN have $[40 \times d, 1024, 1024]$ as hidden dimension for each of their layers. Note that in the actor, $d$ = observation dimension whereas in the critic, $d$ = observation dimension + actor dimension.

We build upon DrQv2 (Yarats et al., 2021), and derive conv FFN by replacing the first convolutional layer with a 1x1 convolution using the sine function as non-linearity. We use the initialization scheme described in Equation 14 for the weights and biases.

## A.5   SUMMARY OF VARIOUS FOURIER FEATURES

Using Fourier features to correct the spectral-bias is a general technique that goes beyond a particular parameterization. Hence we present comparison between

- **vanilla MLP** is a stack of $t$ linear layers with ReLU activation

$$\text{MLP}(x) = f^t \circ \text{ReLU} \circ \cdots f^2 \circ \text{ReLU} \circ f^1(x)$$

- **Fourier features network (FFN) (Ours)** uses sine activation with random phase-shift, to replace the first layer of the network with learned Fourier features

$$\text{LFF}(x) = \sin(Wx + c), \quad W_{i,j} \sim \mathcal{N}(0, \pi b/d), c_i \sim \mathcal{U}(-\pi, \pi)$$

so that the Fourier features network (FFN) is

$$\text{FFN}(x) = f^t \circ \text{ReLU} \circ \cdots f^2 \circ \text{LFF}(x)$$

- **RFF (Tancik et al., 2020)** that uses sine and cosine pairs concatenated together

$$\text{RFF}_{\text{Tancik}}(x) = [\sin(2\pi Wx), \cos(2\pi Wx)], \quad W_{i,j} \sim \mathcal{N}(0, \sigma^2)$$

- **SIREN network (Sitzmann et al., 2020)** that stacks learned Fourier layers through our the entire network, using the Sitzmann initialization according to

$$lff(x) = \sin(Wx + c), \quad W_{i,j} \sim \mathcal{U}(-\sqrt{6}/\sqrt{n}, \sqrt{6}/\sqrt{n}), c_i \sim \mathcal{U}(-\pi, \pi)$$

where the t-layer network

$$\text{SIREN}(x) = lff^t \circ \cdots lff^2 \circ lff^1(x).$$

It is critical to note what each layer has a distinct bandwidth scaling parameter. Hence instead of a single scaling hyperparameter $b$ for the random matrices, Siren has a set $\{b_i\}$ that each need to be tuned.

We found that it is important to use isotropic distributions for the weights. This means using Gaussian distributions or orthogonal matrices for the weight matrix. Uniform distribution is anisotropic.

To stack LFF (which becomes SIREN), the network requires a "fan-out" in the latent dimensions to accommodate the successive increase in the number of features. SIREN is difficult to tune because the initialization scheme is incorrect, as it depends on the inverse quadratic root of the input dimension. Correcting the initialization to depend inversely on the input dimension fixes such issue.

### A.6 EFFECT OF BANDWIDTH PARAMETER $b$ ON TOY MDP

The weight distribution in the random Fourier features offers a way to control the generalization (interpolation) in the neural value approximator. In Figure 15, we compare the learned Q function using Fourier feature networks initialized with different band-limit parameter $b$. With larger $b$s, the network recovers sharper details on the optimal Q function. We also provide the result for an FFN with the target network turned off.

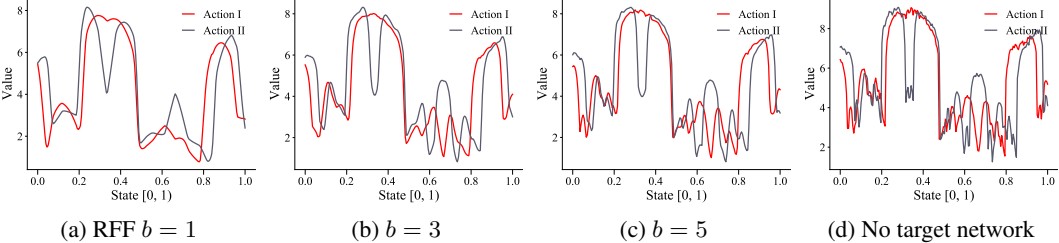

(a) RFF $b = 1$  (b) $b = 3$  (c) $b = 5$  (d) No target network

Figure 15: Q-value approximation on the toy MDP with different bandlimit parameter $b$. (a - c) showing $b = \{1, 3, 5\}$. (d) showing a FFN without target network.

### A.7 ABLATIONS AND SENSITIVITY ANALYSES

**Sensitivity to bandwidth** $b$  Since FFN introduces a bandwidth hyperparameter $b$, it is natural to ask how the choice of b affects FFN's performance. Figure 16 shows that FFN's performance does vary with choice of $b$. Furthermore, the best performing value of $b$ differs with environment. This difference arises from the fact that the optimal value function for different environments have different spectral bias and hence requires different bandwidth $b$ for FFN.

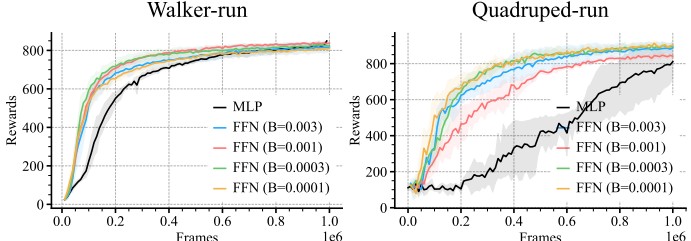

Figure 16: Learning curve with FFN for different values of B on Walker-run and Quadruped-run. We use *SAC* as the RL algorithm. We observe that different B values lead to different performances and hence, we must choose the B value carefully for each environment.

**MLP Actor + FFN Critic**  Experiments we have presented so far use FFN for both the actor and the critic. Our intuition is that the main benefit comes from using FFN in the critic. Results in Figure 17 shows that this is indeed the case – using an MLP for the actor does not affect the learning performance, whereas using an MLP for the critic causes the performance to match those of the MLP baseline.

**Sensitivity to (Fourier) Feature-Input Ratio**  In addition to the bandwidth $b$, we need to choose the Fourier feature dimension $D$ for FFN. We maintained the feature-input ratio $\frac{D}{d}$ to be $40$. But can we get away with a lower feature-input ratio? Figure 18 shows that it is important to maintain feature-input ratio to be at least $40$ and any reduction in the feature-input ratio hurts the performance of FFN.

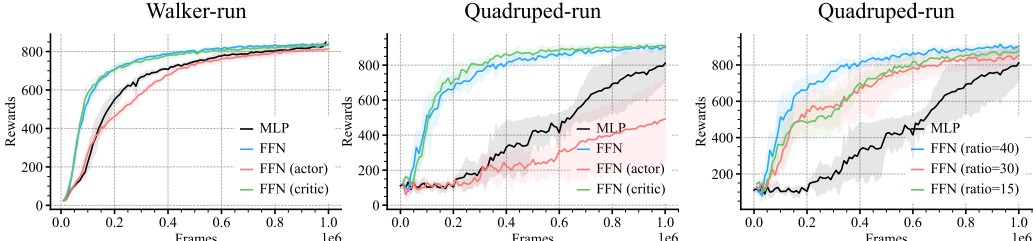

Figure 17: Using FFN for critic is as good as using FFN for both actor and critic. However, using learned FFN only for actor is similar to the MLP baseline. This indicates the gain mainly comes from better value approximation.

Figure 18: Learning curve with different Fourier dimension ratio on Quadruped-run. Lowering the ratio decreases the performance.

## A.8  FULL RESULTS WITH SAC

We include the full set of results on eight DMC domains in Figure 19 using soft actor-critic (Haarnoja et al., 2018).

## A.9  FULL RESULTS WITH DDPG

The benefits FFN brings generalize beyond soft actor-critic to other RL algorithms. In this section we present results based on deep deterministic policy gradient (DDPG, see Lillicrap et al. 2015). Figure 20 shows learning curve of the FFN vs the MLP on eight DMC domains.

## A.10  REDUCING VALUE APPROXIMATION ERROR

We include the full result on value approximation error on eight DMC domains in Figure 21. FFN consistently reduces the value approximation error *w.r.t* the MLP baseline especially at earlier stages of training when insufficient amount of optimization has occurred.

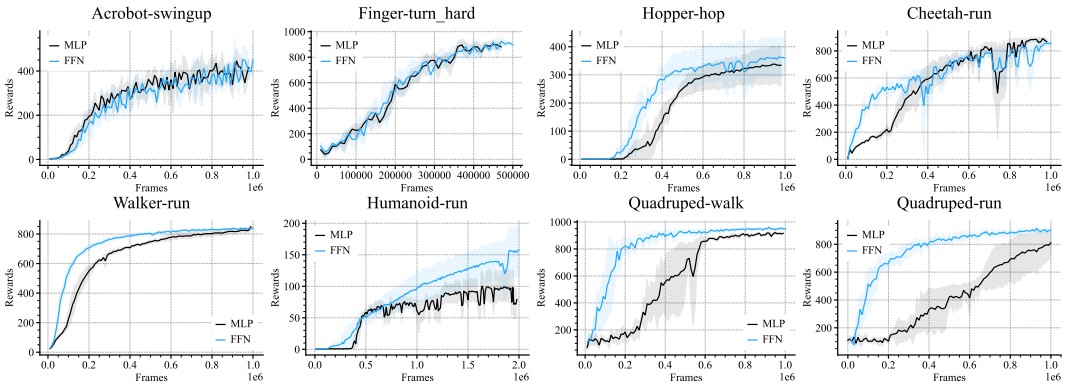

Figure 19: Learning curve with FFN applied to *SAC* on the DeepMind control suite. Domains are ordered by input dimension, showing an overall trend where domains with higher state dimension benefits more from Fourier features. We use a (fourier) feature-input ratio of $40 : 1$.

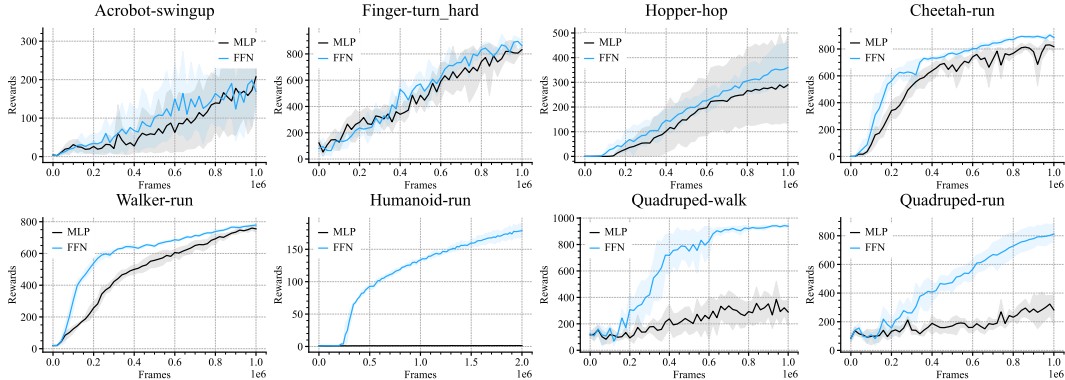

Figure 20: Learning curve with FFN applied to *DDPG* on the DeepMind control suite. Domains are ordered by input dimension. The over trend agrees with results on soft actor-critic, where domains with higher state dimension benefits more from random Fourier features. The same feature-input ratio of $1 : 40$ is applied.

## A.11  MEASURING PARAMETER CHANGES IN FFN AND MLP

We can visualize the amount of changes the FFN and the MLP experience in the *weight space*. Let $\mathcal{W}_t^k$ and $\mathcal{B}_t^k$ refer to the flattened weight and bias vectors from layer $k$ in the critic after training over $t$ environment frames which corresponds to the number of optimization steps times a multiple. Figure 22 shows the evolution of $\|\mathcal{W}_t - \mathcal{W}_0\|$ for all eight domains. Figure 23 shows the evolution of $\|\mathcal{B}_t - \mathcal{B}_0\|$. Both collected with soft actor-critic.

## A.12  REMOVING THE TARGET NETWORK

We present the full result without the target network in Figure 24. While MLP completely flat-lines in all environments, FFN matches its original performance with target networks in Finger-turn, and Quadruped-walk. It suffers performance loss but still manages to learn in Cheetah-run, Walker-run, and Quadruped-run. Hopper-hop, Humanoid-run, and Acrobat-swing up are difficult for model-free algorithms, and FFN fails to learn without target network.

We can inspect the value approximation error. Without a target network, both MLP and FFN do poorly on Hopper-hop, so the result on this domain is uninformative. In Walker-run, removing the target network reduces the value approximation error even further. In Quadruped walk, FFN without target network perform worse than FFN with target network, and has slightly higher approximation error. Both FFN baselines however have lower approximation errors than the MLP. In Quadruped-run without the target network, the approximation error diverges to large value, then converges back. The source of this divergence remains unclear.

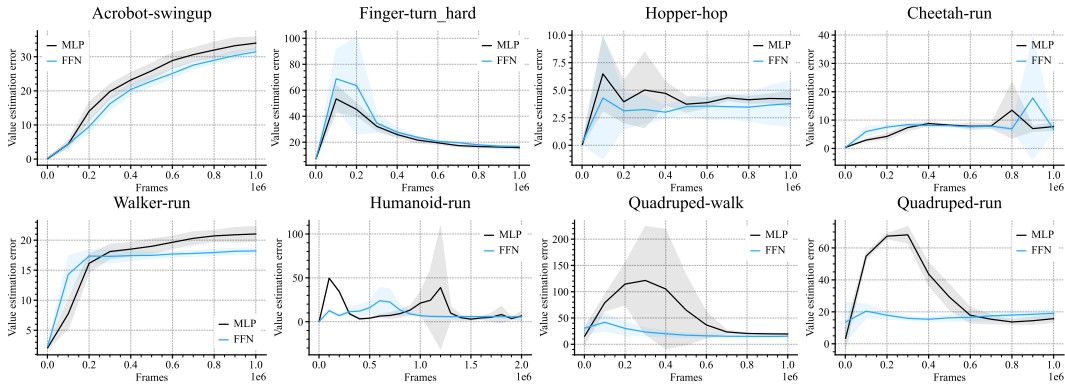

Figure 21: Value approximation error with *SAC* on the DeepMind control suite. FFN reduces the error *w.r.t* the MLP, especially during earlier stages of learning where the gradient updates is low.

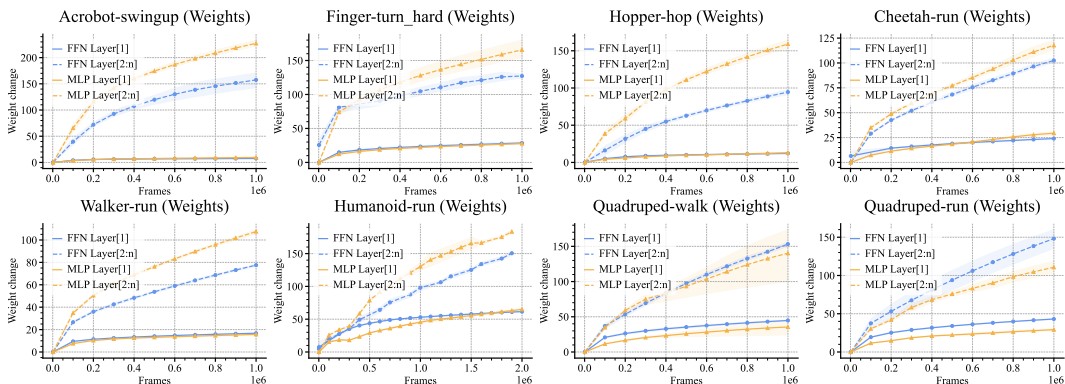

Figure 22: *Weight change* in FFN and MLP during training of RL agents with *SAC* on the DeepMind control suite. The results are mixed and FFN's weight parameters undergo less change than MLP's weight parameters only in some environments.

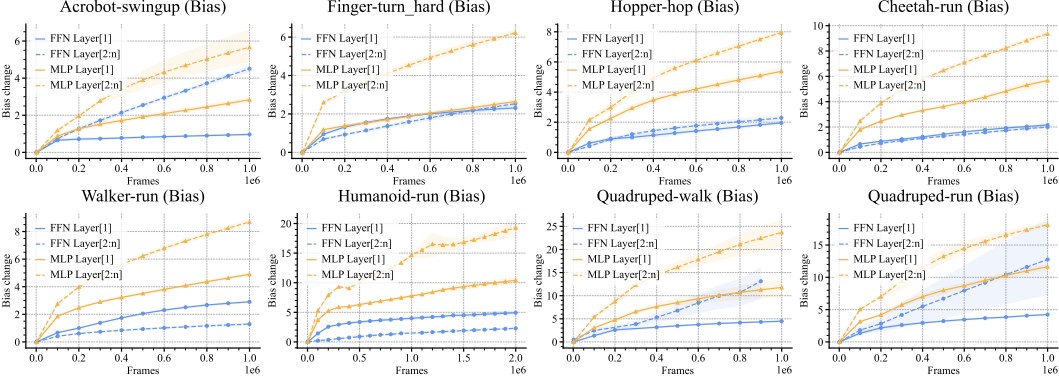

Figure 23: *Bias change* in FFN and MLP during training of RL agents with *SAC* on the DeepMind control suite. Given FFN's bias parameters have better initialization, they undergo less change than MLP's bias parameters.

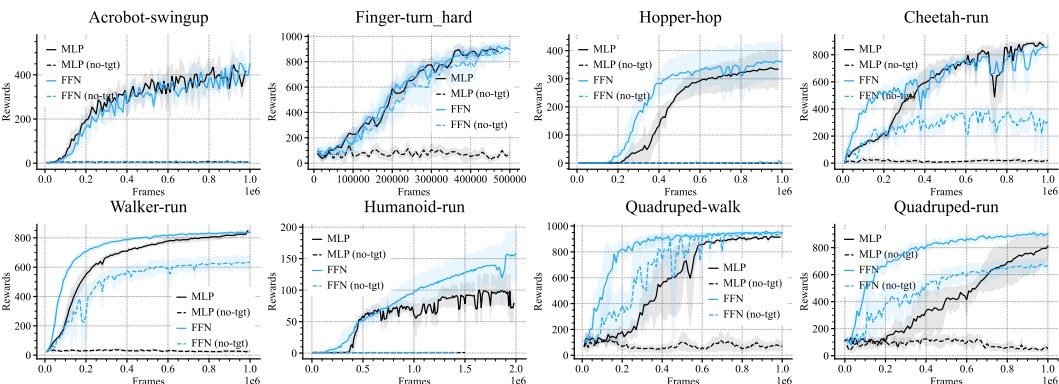

Figure 24: Since FFN require fewer gradients for value function estimation, its performance doesn't degrade as much when target value networks are removed. On the contrary, MLP completely fails when target value networks are removed.

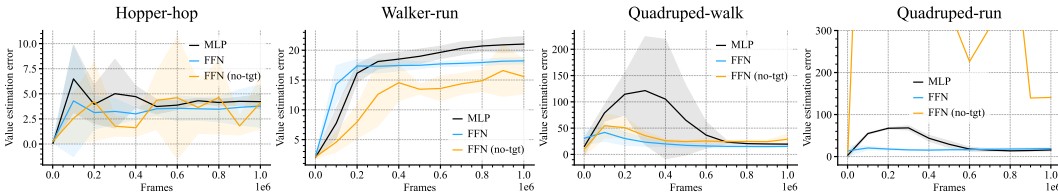

Figure 25: Value approximation error with FFN vs MLP using *SAC*. The divergence is especially prominent at the beginning of learning when the data is sparse. FFN reduces the variance in these regimes by making more appropriate bias-variance trade-off than the MLP baseline.

