# OpenReview forum: "Overcoming The Spectral Bias of Neural Value Approximation"
_ICLR.cc/2022/Conference — ICLR 2022 Poster_

### Official Review · Reviewer_fVzS · 2021-10-31

**Correctness:** 4
**Technical Novelty And Significance:** 2
**Empirical Novelty And Significance:** 4
**Recommendation:** 6
**Confidence:** 4

**Main Review:**

Pros:

The motivation is clear and sound. The problem discussed is well-defined and the introduction to the problem makes sense.

The proposed method RFN was shown to be effective on both toy tasks and high-dimensional robotic control tasks with DDPG. In particular, RFN +DDPG show a significant performance gain in the challenging humanoid control task. Also, RFN can theoretically be applied to any neural network for function approximation.

Concerns / Questions:

It is impressive to see DDPG+RFN works on Humanoid in contrary with the original DDPG’s terrible performance. However, we also know that more advance algorithms like SAC or TD3 can also get much better performance than DDPG (similar to RFAC) without resolving the spectral bias issue. Does this mean that spectral bias is not critical to performance if other aspects of the RL algorithm are good?

While the empirical results are good, the novelty is a bit weak. My understanding is that the authors borrowed the RFN from recent studies and made an investigation of the hyper-parameters so as to fit in RL. Please correct me if I have any misunderstanding.

For the RFF layer, it was written that “We find this initialization scheme work well across all domains, with minimum requirement on tuning, even when stacked as learnable units.”. I am a bit confused here. Are the weights and bias of RFF layer learnable or not in your implementations?

The format of references is not professional nor consistent. Some of them has hyperlinks and some not. Also, Please carefully confirm the published venue of each citation. For example, the Adam paper was published on ICLR, please do not use the arxiv version.

For the MuJoCo experiments, I would like to see RFAC’s performance on other tasks such as hopper, walker2D, etc., if possible. These results may give people more understanding of RFAC’s effectiveness on (relatively) simpler and lower-dimensional control tasks.

For the performance curves (figure 8-12). How many random seeds were tested? What was the shaded area (STD of S.E.M.?) In figure 9, the authors claimed that “its performance varies on Humanoid with different values of b”. But from the plot I cannot see clear statistically significant difference.


Smaller problems:
1.	The abbreviation TD needs to be explained at the first usage.
2.	"Note that DDPGtakes 1 gradient step" -> DDPG takes
3.	It is a bit weird to put Related Works in the end. How about putting it before the Discussion?
4.	Comma and period should be properly used in the equations.


**Summary Of The Paper:**

The paper discussed the problem of spectral bias in MLPs (i.e., dependence on the higher-frequency part of input feature is under-fitted) within RL domain. The authors first described the problem with toy examples, which clearly explained this problem. Then it was suggested to use random Fourier feature network (RFN) to transform the input to frequency domain and then feed to the network, to resolve this issue.

Experiments demonstrated effectiveness of RFN to learn a complicated Q-function in low-dimensional state space. And it was also shown that RFN contribute to performance gain with DDPG on high-dimensional robotic tasks. The authors also did an “ablation study” (I would rather call it “sensitivity analysis” instead) to empirically show how the performance depends on the choice of hyper-parameters.


**Summary Of The Review:**

The paper discussed an important problem in RL that was previously under-investigated. Overall, I think the paper successfully addressed it by suggesting the usage of a random Fourier feature network. However, the novelty is limited since the main contribution borrows the methodology in existing papers, together with the aforementioned concerns, I am not convinced that the paper meets the acceptance criteria.

### Post-rebuttal
I really appreciate the authors' work for providing new results on DMC tasks. The empirical results are much more convincing and comprehensive now. I would like to raised up my score because I think the current results have successfully supported the motivations. However, now the only and the largest concern is that necessity of a new round of review for such a major revision (unfortunately this is not possible now). After all, I set my recommendations to weak accept since I cannot (and should not) access the newest revised version

---

> ### Author Response · Authors · 2021-11-23
> **Author's reply (1/2)**
>
> We thank reviewer fVzS for their constructive feedback, and would like to address specific point below:
>
> > While the empirical results are good, the novelty is a bit weak. My understanding is that the authors borrowed the RFN from recent studies and made an investigation of the hyper-parameters so as to fit in RL. Please correct me if I have any misunderstanding.
>
> The main idea in this work is not to introduce a novel deep RL architecture. Rather, it is to make a connection between recent discoveries by the graphics community to the longstanding problem of value function underfitting in deep RL.
>
> Value approximation is a key issue in deep reinforcement learning that has inspired a number of high-impact works [^hesselt] [^fujimoto]. All of the aforementioned work however, tackle the issue from the perspective of better learning procedures while using generic multi-layer perceptron networks. There exists a blind spot around how we parameterize the neural value approximator.
>
> Equiped with new tools, we want to make a contribution to the field of deep reinforcement learning by pointing out that the issue arises from the tension between the learning objective of trying to fit a complex value function, and the simplicity bias intrinsic to the neural network itself. **We do so through a novel, spectral perspective (section 3),** and offer a simple fix that is motivated by recent developments from two entirely different communities. These recent development from graphicsl community are quite new [^tanick] [^siren] by themselves. And as we have shown, applying these kernels to RL poses its own challenges an required additional care in ways we initialize the features.
>
> For these reasons, we would like to argue that our work makes an interesting connection between new developments in deep learning theory and graphics with a fundational problem in deep RL. We also offer a general class of fixes that are simple to implement and can scale up to complex problems. Our key contribution is not a specific parameterization, but the spectral and neural tangent kernel perspective on value approximation with neural networks.
>
> The authors appreciate your concerns and thoughful feedback. And would like to address them in detail below:
>
> [^hesselt]: Hasselt, H. (2010) ‘Double Q-learning’, in Lafferty, J. et al. (eds) Advances in Neural Information Processing Systems. Curran Associates, Inc.
>
> [^fujimoto]: Fujimoto, S., van Hoof, H. and Meger, D. (2018) ‘Addressing Function Approximation Error in Actor-Critic Methods’, in Dy, J. and Krause, A. (eds) Proceedings of the 35th International Conference on Machine Learning. PMLR (Proceedings of Machine Learning Research), pp. 1587–1596.
>
> [^tanick]: Tancik, M., Srinivasan, P. P. and Mildenhall, B. (2020) ‘Fourier features let networks learn high frequency functions in low dimensional domains’
>
> [^siren]: Sitzmann, V. et al. (2020) ‘Implicit Neural Representations with Periodic Activation Functions’, arXiv [cs.CV].
>
> [^ntk]: Jacot, A., Gabriel, F. and Hongler, C. (2018) ‘Neural Tangent Kernel: Convergence and Generalization in Neural Networks’, arXiv [cs.LG]. Available at: http://arxiv.org/abs/1806.07572.
>
> ----
>
> ## Additional Actor-critic Baselines and DMC Domains
>
> > It is impressive to see DDPG+RFN works on Humanoid in contrary with the original DDPG’s terrible performance. However, we also know that more advance algorithms like SAC or TD3 can also get much better performance than DDPG (similar to RFAC) without resolving the spectral bias issue. Does this mean that spectral bias is not critical to performance if other aspects of the RL algorithm are good?
>
> SAC and TD3 are largely identical when it comes to the techniques they introduced to address over-estimation. These methods are largely orthogonal our work.
>
> Spectral bias is still an issue in these TD-learning-based reinforcement learning algorithms. To show this, **we have included new experimental results with the state-of-the-art TD3 learning algorithm** in section 2 of the appendix. We report results on the DMC Hopper domain in this updated draft, but are in the process of scaling up to common DMC domains.
>
> ## Comparing Between Learned Vs Fixed Random Fourier Features
>
> In appendix 2, we include additional comparison between the following kernels:
> 1. vanilla MLP
> 2. Random Fourier Features (ours) that uses sine activation with random phase-shift
> 3. RFF (Tanick) that uses sine and cosine pairs
> 4. LFF (learned fourier features) (ours)
> 5. Siren network that uses stacked learned Fourier layers and Sitzmann initialization
>
> We offer these results on Ant, Cheetah, and Humaoid domains. Sirens in particular works poorly, which agrees with the observations that the network performance is quite sensitive to the weight initialization.

---

> > ### Author Response · Authors · 2021-11-23
> > **Author's reply (2/2)**
> >
> >
> > ## New Results on The Effective Rank
> >
> > We were looking for a quantitative way to show that random Fourier features improve the complexity of the learned value function. We do so in a new experiment by computing the effective rank [^roy_2007] of the penmultimate layer of the value function over data in the replay buffer. Our result (section A.3 in the appendix) indicate value functions equipped with the random fourier features indeed learn a higher rank solution, beside obtaining superior perfomance.
> >
> > | Q function rank | RFAC            | DDPG            |
> > | --------------- | --------------- | --------------- |
> > | Ant             | $2.86 \pm 1.32$ | $1.08\pm 1.07$  |
> > | Humanoid        | $2.94 \pm 1.26$ | $1.49 \pm 1.22$ |
> > | Half Cheetah    | $2.94 \pm 1.12$ | $1.01 \pm 1.00$ |
> >
> > [^roy_2007]: Roy, O. and Vetterli, M. (2007) ‘The effective rank: A measure of effective dimensionality’, in 2007 15th European Signal Processing Conference, pp. 606–610.

---

> > > ### Comment · Reviewer_fVzS · 2021-11-24
> > > **Incomplete rebuttal?**
> > >
> > > I appreciate the authors for the reply. However, the rebuttal is incomplete, which only replied to the first 2 of my total 6 concerns. If the authors are still working on the remaining parts, I will wait and will be happy to raise my score if the complete rebuttal is convincible. Otherwise, I keep my current evaluation.

---

> > > > ### Author Response · Authors · 2021-11-25
> > > > **Re: Incomplete rebuttal**
> > > >
> > > > We are sorry for not addressing all your concerns in our first response. We were working on formulating our responses, and they are below:
> > > >
> > > > ## Concern #3: Is the RFF trained?
> > > >
> > > > > For the RFF layer, it was written that “We find this initialization scheme work well across all domains, with minimum requirement on tuning, even when stacked as learnable units.”. I am a bit confused here. Are the weights and bias of RFF layer learnable or not in your implementations?
> > > >
> > > > The weights and biases are not learned in the RFF layer. We refer to the learned variant as "LFF", or "learned Fourier features". In our experiment, we found out that the weight and bias parameters in the learned variant, LFF, "oscillate" in a narrow range around the initialization. Specifically, weights changed by ~[1e-6, 1e-4].
> > > >
> > > > Tanick et al also report that the NTK of the Random Fourier Feature network does not change much if they train those. This is in contrast to the learning dynamics of a linear layer with ReLU activation, where the magnitude of the parameters will change significantly. Now because gradient-based updated does not affect the overall distribution of the weight parameters in the random Fourier layer, one can not expect gradient updates to fix a bad initialization of random Fourier layer. We therefore choose to keep RFF fixed, instead of making it learnable.
> > > >
> > > > ## Concern #5: RFF Effectiveness on Hopper, Walker2D
> > > >
> > > > > For the MuJoCo experiments, I would like to see RFAC’s performance on other tasks such as hopper, walker2D, etc., if possible. These results may give people more understanding of RFAC’s effectiveness on (relatively) simpler and lower-dimensional control tasks.
> > > >
> > > > We now include results on Hopper in the updated draft (Appendix A.4). We are in process of running experiments on more dmc domains and would post the results as soon as we get it.
> > > >
> > > >
> > > > ## Concern #6: Experiment Details and Statistical Significance
> > > >
> > > > >  For the performance curves (figure 8-12). How many random seeds were tested? What was the shaded area (STD of S.E.M.?) In figure 9, the authors claimed that “its performance varies on Humanoid with different values of b”. But from the plot I cannot see clear statistically significant difference.
> > > >
> > > > We ran 5 random seeds across all experiments. The shaded area corresponds to 1 standard deviation (by lower 15.9 and 84.1 percentile) of the mean reward.
> > > >
> > > > Our sensitivity test on HalfCheetah and Ant shows that the performance of random Fourier actor-critic is largely insensitive to the bandwidth parameter $b$, where the RFAC performance with different $b$ fall within the error range of each other. However, we wanted to point out that while $b=5, 15, 20$ are grouped together, $b=10$ is an outlier that performed slightly worse.
> > > >
> > > > We have updated the draft on our end to make this more clear.
> > > >
> > > > ## Concern #4: Reference Format
> > > >
> > > > > The format of references is not professional nor consistent. Some of them has hyperlinks and some not. Also, Please carefully confirm the published venue of each citation. For example, the Adam paper was published on ICLR, please do not use the arxiv version.
> > > >
> > > > We applogize for this negligence. We have gone through the references to:
> > > > 1. use a consistent format from Google Scholar
> > > > 2. remove the hyperlinks
> > > > 3. found the published venues to the best of our abilities
> > > >
> > > > There are two remaining formatting issues in the uploaded draft (Arpit2017 and the MountainCar software). We have fixed these on our end.

---

> > > > > ### Comment · Reviewer_fVzS · 2021-11-26
> > > > > **Reviewer response**
> > > > >
> > > > > Thanks for the reply! As the authors have addressed most of my concerns, I would like to increase my score. I have some more comments.
> > > > >
> > > > > Now I feel this paper is at acceptance borderline. The pros are that the being discussed spectral bias problem is an important but previously under-explored topic and the being proposed method is effective on some of the difficult robotic control tasks. I believe that the related results should be reported to the RL community for better understanding approximation errors/biases of deep RL.
> > > > >
> > > > > However, it is because this topic (spetral bias in value approximation) is important, the takeaways should be more carefully and throughoutly discussed. For example,
> > > > >
> > > > > 1. In Fig. 14, it is written that "*RFF network has lower variance, and
> > > > > out-performs the vanilla MLP baseline on average.*".  It is wrong to say "outpeform ... on average" when the two sets of samples have no significant difference because the "average" is not true average, but just sampled average. The statistical significance should be more carefully addressed [1] [2].
> > > > >
> > > > > 2. While the method is shown to be effective on Ant and Humanoid (high-dimensional actions, Fig. 9 and 11), it does not outperform baseline (DDPG+DrQv2) on Hopper and Half-Cheetah (low-dimensional actions, Fig.9, 11 and 14).  I think there should be more discussion about "In what kind of tasks we should use RFAC" .
> > > > >
> > > > > 3. What are the takeaways from the results of effective rank? e.g., the results show using RFF can largely increase effective rank of Q-function for all HalfCheetah, Ant, and Humanoid. This is expected, but what is the implications for RL? For HalfCheetah, (Fig.9) the RL performance of RFAC does not significantly outperforms baseline, how is this result related to effective rank (i.e. why effective rank of HalfCheetah is largely increased by RFAC but RL performance of HalfCheetah  is not)?
> > > > >
> > > > > Also, I think there is a need to re-organize the experimental results (as the authors are also testing more environments, which I think is very good to include them with a major revision). For instance, Fig. 14 should be of the same importance as Fig. 9/11, so it should be presented in the main texts rather than in the appendix.
> > > > >
> > > > > Overall, I will clearly support this work to be published on top-tier venues like ICLR if the presentation of reults are improved. I believe the quality of the current manuscript can be largely increased by re-organizing the structure and complement the discussion of results by including more environment.
> > > > >
> > > > > ### Ref:
> > > > >
> > > > > [1] Colas C, Sigaud O, Oudeyer P Y. A Hitchhiker's Guide to Statistical Comparisons of Reinforcement Learning Algorithms[J]. arXiv preprint arXiv:1904.06979, 2019.
> > > > >
> > > > > [2] Henderson P, Islam R, Bachman P, et al. Deep reinforcement learning that matters[C]//Proceedings of the AAAI conference on artificial intelligence. 2018, 32(1).

---

> > > > > > ### Author Response · Authors · 2021-11-28
> > > > > > **Author's reply: Addressing new concerns and results on DMC domains (1/3)**
> > > > > >
> > > > > > We thank reviewer for the comment and we will now address these concerns below:
> > > > > >
> > > > > > ## Restructuring and New Results
> > > > > >
> > > > > > > Also, I think there is a need to re-organize the experimental results (as the authors are also testing more environments, which I think is very good to include them with a major revision). For instance, Fig. 14 should be of the same importance as Fig. 9/11, so it should be presented in the main texts rather than in the appendix.
> > > > > >
> > > > > > Thank you for this suggestion! The authors have been running experiments non-stop, and now include RFF applied to both SAC and DDPG on **eight DeepMind Control tasks**. This is in addition to those we already include in the appendix. We agree with your suggestion that to accommodate these additional results, the results section needs to be restructured. Here is the structure that we plan to have:
> > > > > >
> > > > > > ```markdown
> > > > > > **Experiments**
> > > > > > 1. Fitted Q Iteration with Mountain Car
> > > > > > 2. Continuous Control Problem Setup
> > > > > >   a. Environments: MuJoCo and DeepMind Control suite
> > > > > >   b. Actor-critic Baselines
> > > > > >   c. Random Fourier Feature Variants and Implementation
> > > > > > 3. Continuous Control Performace
> > > > > > 	a. MuJoCo Domains: DDPG and TD3  **Moved from appendix 5**
> > > > > >   b. DeepMind Control suite: DDPG and SAC  **Moved from appendix 4**
> > > > > > 4. Analysis
> > > > > > 	a. Measuring The Complexity of Learned Value Functions **Moved from Appendix 3**
> > > > > >   b. Sensitivity to Bandwidth Parameter **New + Appendix 1**
> > > > > >   c. Latent Dimension vs Performance
> > > > > >
> > > > > > **Appendix**
> > > > > > 1. Toy MDP Experimental Details
> > > > > > 2. Fourier Kernel Pseudo-code
> > > > > > 3. Network Architectures
> > > > > > 4. Full Results on Sensitivity to Bandwidth
> > > > > > 	a. DDPG on DMC
> > > > > >   b. SAC on DMC
> > > > > > ```
> > > > > >
> > > > > >
> > > > > > ---
> > > > > > ## New, Comprehensive Results on DeepMind Control
> > > > > >
> > > > > > We reproduce our new result on **eight DeepMind Control suite domains** in table form below, where we order the domains by their state-action space complexity:
> > > > > >
> > > > > > **Fourier Features on DDPG and DeepMind Control Tasks**
> > > > > >
> > > > > > | Name             | Observation   | Action  | DDPG + MLP        | DDPG + RFF        |
> > > > > > | ---------------- | ------------- | ------- | ----------------- | ----------------- |
> > > > > > | Acrobot-swingup  | Box(6,)       | Box(1,) | $174.3 \pm 92.6$  | $172.9 \pm 108.2$ |
> > > > > > | Finger-turn_hard | Box(12,)      | Box(2,) | $816.8 \pm 92.1$  | $864.1 \pm 53.8$  |
> > > > > > | Hopper-hop       | Box(15,)      | Box(4,) | $282.1 \pm 189.9$ | $353.1 \pm 106.9$ |
> > > > > > | Cheetah-run      | Box(17,)      | Box(6,) | $787.2 \pm 77.9$  | $\boldsymbol{890.5 \pm 18.9}$ |
> > > > > > | Walker-run       | Box(24,)      | Box(6,) | $750.9 \pm 17.9$  | $\boldsymbol{772.1 \pm 11.5}$ |
> > > > > > | Humanoid-run     | Box(67,)      | Box(21,)| $1.3 \pm 0.1$     | $\boldsymbol{177.9 \pm 14.7}$ |
> > > > > > | Quadruped-run    | Box(78,)      | Box(12,)| $290.9 \pm 119.3$ | $\boldsymbol{799.4 \pm 75.9}$ |
> > > > > > | Quadruped-walk   | Box(78,)      | Box(12,)| $311.1 \pm 143.4$ | $\boldsymbol{939.6 \pm 12.3}$ |
> > > > > >
> > > > > > **Fourier Features on SAC and DeepMind Control Tasks**
> > > > > >
> > > > > > | Name             | Observation   | Action  | SAC + MLP        | SAC + RFF        |
> > > > > > | ---------------- | ------------- | ------- | ----------------- | ----------------- |
> > > > > > | Acrobot-swingup  | Box(6,)       | Box(1,) | $101.7 \pm 87.2$  | $\boldsymbol{219.8 \pm 61.4}$ |
> > > > > > | Finger-turn_hard | Box(12,)      | Box(2,) | $836.2 \pm 80.8$  | $552.2 \pm 182.0$ |
> > > > > > | Hopper-hop       | Box(15,)      | Box(4,) | $147.8 \pm 88.9$  | $\boldsymbol{322.8 \pm 27.8}$ |
> > > > > > | Cheetah-run      | Box(17,)      | Box(6,) | $829.3 \pm 59.8$  | $881.0 \pm 12.9$ |
> > > > > > | Walker-run       | Box(24,)      | Box(6,) | $852.4 \pm 7.1$   | $811.7 \pm 22.1$  |
> > > > > > | Humanoid-run     | Box(67,)      | Box(21,)| $191.5 \pm 40.3$  | $\boldsymbol{261.6 \pm 35.4}$ |
> > > > > > | Quadruped-run    | Box(78,)      | Box(12,)| $582.2 \pm 216.9$ | $\boldsymbol{919.4 \pm 20.4}$ |
> > > > > > | Quadruped-walk   | Box(78,)      | Box(12,)| $485.1 \pm 280.5$ | $\boldsymbol{944.6 \pm 14.1}$ |
> > > > > >
> > > > > > ## Images of the Learning Curves
> > > > > >
> > > > > > We attach updated learning curves below:
> > > > > > - DDPG on DeepMind Control: https://imgur.com/a/OhcwI7A
> > > > > > - SAC on DeepMind Control: https://imgur.com/a/yivWdTj

---

> > > > > > > ### Author Response · Authors · 2021-11-28
> > > > > > > **Author's reply: Addressing new concerns and results on DMC domains (2/3)**
> > > > > > >
> > > > > > > ## Discussion on Larger Gains in More Complex Domains
> > > > > > >
> > > > > > > > While the method is shown to be effective on Ant and Humanoid (high-dimensional actions, Fig. 9 and 11), it does not outperform baseline (DDPG+DrQv2) on Hopper and Half-Cheetah (low-dimensional actions, Fig.9, 11 and 14).
> > > > > > >
> > > > > > > To answer this question, it is best to arrange the domains in the order of the dimensionality of the observation and action space. In the table below, we notice an overall trend where complex domains benefit much more significantly from random Fourier feature than simpler domains.
> > > > > > >
> > > > > > > This indicates that value approximation due to the spectral bias might not be a bottleneck on these simpler domains that have lower dimensions, and the performance bottleneck lies somewhere else.
> > > > > > >
> > > > > > > | Name             | Observation   | Action  | DDPG + MLP        | DDPG + RFF        |
> > > > > > > | ---------------- | ------------- | ------- | ----------------- | ----------------- |
> > > > > > > | Acrobot-swingup  | Box(6,)       | Box(1,) | $174.3 \pm 92.6$  | $172.9 \pm 108.2$ |
> > > > > > > | Finger-turn_hard | Box(12,)      | Box(2,) | $816.8 \pm 92.1$  | $864.1 \pm 53.8$  |
> > > > > > > | Hopper-hop       | Box(15,)      | Box(4,) | $282.1 \pm 189.9$ | $353.1 \pm 106.9$ |
> > > > > > > | Cheetah-run      | Box(17,)      | Box(6,) | $787.2 \pm 77.9$  | $\boldsymbol{890.5 \pm 18.9}$ |
> > > > > > > | Walker-run       | Box(24,)      | Box(6,) | $750.9 \pm 17.9$  | $\boldsymbol{772.1 \pm 11.5}$ |
> > > > > > > | Humanoid-run     | Box(67,)      | Box(21,)| $1.3 \pm 0.1$     | $\boldsymbol{177.9 \pm 14.7}$ |
> > > > > > > | Quadruped-run    | Box(78,)      | Box(12,)| $290.9 \pm 119.3$ | $\boldsymbol{799.4 \pm 75.9}$ |
> > > > > > > | Quadruped-walk   | Box(78,)      | Box(12,)| $311.1 \pm 143.4$ | $\boldsymbol{939.6 \pm 12.3}$ |
> > > > > > >
> > > > > > > | Name             | Observation   | Action  | SAC + MLP        | SAC + RFF        |
> > > > > > > | ---------------- | ------------- | ------- | ----------------- | ----------------- |
> > > > > > > | Acrobot-swingup  | Box(6,)       | Box(1,) | $101.7 \pm 87.2$  | $\boldsymbol{219.8 \pm 61.4}$ |
> > > > > > > | Finger-turn_hard | Box(12,)      | Box(2,) | $\boldsymbol{836.2 \pm 80.8}$  | $552.2 \pm 182.0$ |
> > > > > > > | Hopper-hop       | Box(15,)      | Box(4,) | $147.8 \pm 88.9$  | $\boldsymbol{322.8 \pm 27.8}$ |
> > > > > > > | Cheetah-run      | Box(17,)      | Box(6,) | $829.3 \pm 59.8$  | $\boldsymbol{881.0 \pm 12.9}$ |
> > > > > > > | Walker-run       | Box(24,)      | Box(6,) | $\boldsymbol{852.4 \pm 7.1}$   | $811.7 \pm 22.1$  |
> > > > > > > | Humanoid-run     | Box(67,)      | Box(21,)| $191.5 \pm 40.3$  | $\boldsymbol{261.6 \pm 35.4}$ |
> > > > > > > | Quadruped-run    | Box(78,)      | Box(12,)| $582.2 \pm 216.9$ | $\boldsymbol{919.4 \pm 20.4}$ |
> > > > > > > | Quadruped-walk   | Box(78,)      | Box(12,)| $485.1 \pm 280.5$ | $\boldsymbol{944.6 \pm 14.1}$ |
> > > > > > >
> > > > > > > ## Where shall we use RFF?
> > > > > > >
> > > > > > > > I think there should be more discussion about "In what kind of tasks we should use RFAC".
> > > > > > >
> > > > > > > We believe one should almost always use random Fourier features because very little tunning is needed, and it allows networks to fit better with less compute. We demonstrate robustness against the bandwidth parameter $b$ through our sensitivity analysis, and show that the policy performance is largely insensitive to $b$ within a large, 2-octave range. We demonstrate the compute efficiency in our motivating example, where an RFF network is able to achieve better fit after 400 gradient steps, than an MLP at 2000 gradient steps. **RFF networks is in general a better function approximator class than vanilla MLP with no obvious downsides.**
> > > > > > >
> > > > > > > At the same time, overfitting could be an issue. For instance SAC+MLP performs better than SAC+RFF in Finger-turn.
> > > > > > >
> > > > > > > ## Effective Rank Takeaways
> > > > > > >
> > > > > > > > What are the takeaways from the results of effective rank?
> > > > > > >
> > > > > > > We hypothesize that value function approximation due to the spectral bias might not be a bottleneck on these simpler domains, and a low-complexity value approximation is sufficient in producing good policies.

---

> > > > > > > > ### Author Response · Authors · 2021-11-28
> > > > > > > > **Author's reply: Addressing new concerns and results on DMC domains (3/3)**
> > > > > > > >
> > > > > > > > ## Statistical Significance of Performance Gains
> > > > > > > >
> > > > > > > > > In Fig. 14, it is written that "RFF network has a lower variance, and out-performs the vanilla MLP baseline on average.". It is wrong to say "outperform ... on average" when the two sets of samples have no significant difference because the "average" is not a true average, but just sampled average. The statistical significance should be more carefully addressed [1][2].
> > > > > > > >
> > > > > > > > Thanks for your comment and we stand corrected. To correctly compute the statistical significance, we now follow the recommendation from Colas *et al* 2019 and use Welch's t-test.
> > > > > > > >
> > > > > > > > We include the p-value as an additional column in the tables below.
> > > > > > > >
> > > > > > > > **DDPG Results on DeepMind Control Suite**
> > > > > > > >
> > > > > > > > | Name             | DDPG + MLP        | DDPG + RFF                    | p_value  |
> > > > > > > > | ---------------- | ----------------- | ----------------------------- |----------|
> > > > > > > > | Acrobot-swingup  | $174.3 \pm 92.6$  | $172.9 \pm 108.2$             | 0.98     |
> > > > > > > > | Finger-turn_hard | $816.8 \pm 92.1$  | $864.1 \pm 53.8$              | 0.28     |
> > > > > > > > | Hopper-hop       | $282.1 \pm 189.9$ | $353.1 \pm 106.9$             | 0.49     |
> > > > > > > > | Cheetah-run      | $787.2 \pm 77.9$  | $\boldsymbol{890.5 \pm 18.9}$ | 0.0076   |
> > > > > > > > | Walker-run       | $750.9 \pm 17.9$  | $\boldsymbol{772.1 \pm 11.5}$ | 0.04     |
> > > > > > > > | Humanoid-run     | $1.3 \pm 0.1$     | $\boldsymbol{177.9 \pm 14.7}$ | 1.11e-5  |
> > > > > > > > | Quadruped-run    | $290.9 \pm 119.3$ | $\boldsymbol{799.4 \pm 75.9}$ | 5.78e-5  |
> > > > > > > > | Quadruped-walk   | $311.1 \pm 143.4$ | $\boldsymbol{939.6 \pm 12.3}$ | 0.00039  |
> > > > > > > >
> > > > > > > >
> > > > > > > > **SAC Results on DeepMind Control Suite**
> > > > > > > >
> > > > > > > > | Name             | SAC + MLP                     | SAC + RFF                     | p_value |
> > > > > > > > | ---------------- | -----------------             | ----------------------------- | ------- |
> > > > > > > > | Acrobot-swingup  | $101.7 \pm 87.2$              | $\boldsymbol{219.8 \pm 61.4}$ | 0.03    |
> > > > > > > > | Finger-turn_hard | $\boldsymbol{836.2 \pm 80.8}$ | $552.2 \pm 182.0$             | 0.015   |
> > > > > > > > | Hopper-hop       | $147.8 \pm 88.9$              | $\boldsymbol{322.8 \pm 27.8}$ | 0.0087  |
> > > > > > > > | Cheetah-run      | $829.3 \pm 59.8$              | $\boldsymbol{881.0 \pm 12.9}$  | 0.01    |
> > > > > > > > | Walker-run       | $\boldsymbol{852.4 \pm 7.1}$  | $811.7 \pm 22.1$              | 0.012   |
> > > > > > > > | Humanoid-run     | $191.5 \pm 40.3$              | $\boldsymbol{261.6 \pm 35.4}$ | 0.022   |
> > > > > > > > | Quadruped-run    | $582.2 \pm 216.9$             | $\boldsymbol{919.4 \pm 20.4}$ | 0.023   |
> > > > > > > > | Quadruped-walk   | $485.1 \pm 280.5$             | $\boldsymbol{944.6 \pm 14.1}$ | 0.02    |
> > > > > > > >
> > > > > > > > ---
> > > > > > > >
> > > > > > > > ## Making Fair Comparisons
> > > > > > > >
> > > > > > > > The main point that [^2] is trying to make is that fair comparison across RL algorithms requires careful calibration of key implementation details. We do abide by this principle in this work.
> > > > > > > >
> > > > > > > > All performance gains in this paper are produced by a single line change to the architecture without modifying the base algorithms. We have uploaded our experimental code base as supplementary material for your reference.

---

> ### Comment · Reviewer_fVzS · 2021-11-29
> **Reply to new results on DeepMind control suite**
>
> I really appreciate the authors' work for providing new results on DMC tasks. The empirical results are much more convicing and comprehensive now.
>
> I would like to raised up my score because I think the current results have successfully supported the motivations. However, now the only and the largest concern is that necessity of a new round of review for such a major revision (unfortunately this is not possible now). After all, I  set my recommendations to weak accept since I cannot (and should not) access the newest revised version

---

### Official Review · Reviewer_TezL · 2021-11-01

**Correctness:** 4
**Technical Novelty And Significance:** 3
**Empirical Novelty And Significance:** 2
**Recommendation:** 6
**Confidence:** 4

**Main Review:**

Strengths:
- The paper tackles an important problem and has an exemplary exposition of the problem and the proposed solution.
- Its results suggest that much more could be done in RL to take into account the peculiarities of the functions being estimated to find good architectures (that may depart from their supervised learning counterparts).

Weaknesses:
- This particular parameterization is fairly close to prior work
- The empirical evaluation on "large scale" problems is limited

On the empirical evaluation, I think this is an important point since, as the authors point out, there is no study in this paper of the generalization properties of RFFs. This is particularly important since we know, e.g. from the SIREN paper, that sinusoidal features are particularly good at memorization.
This may be a fundamental limitation of generalization/function approximation. How much can we get out of generalization when high-frequency features are involved? Results on this question would be potentially very impactful--see for example Hooker et al. (this paper is very much orthogonal to the present work, no need to cite it, but is relevant to the general question of high-frequency things).

Characterising Bias in Compressed Models, Sara Hooker, Nyalleng Moorosi, Gregory Clark, Samy Bengio, Emily Denton, 2020.


Comments:
- Be careful when using citations, some are missing parentheses around them or just the year. Inline citations where the citation is an object in the sentence (e.g. "Newton (1687) showed that X") should use \citet, whereas parenthesized citations should use \citep (e.g. "It has been shown that X (Newton 1687)".
- The text legends in Figures 8-11 could be larger
- An "ablation" is when something is removed. It doesn't seem like anything is removed in the first part section 4.3. Rather, different hyperparameters are tested to attempt to understand their impact.


**Summary Of The Paper:**

This paper proposes a novel parameterization for MLPs, Random Fourier Features (RFF), which amount to using a sinusoidal activation, and initializing parameters to capture different frequencies (by initializing $w_{ij}$ with a variance proportional to a bandwidth hyperparameter $b$) and phases ($b_j \sim U(-\pi,\pi)$).
This particular parameterization is motivated by the need, in RL, to learn value functions which have been found to potentially require high frequency components (whereas vanilla MLPs have a low-frequency/simplicity bias). The paper offers a very detailed exposition of why this is the case and of why RFFs can help.
RFFs are then used on 3 standard control problems, showing some improvement when using DDPG with (vs without) RFFs. The choice of bandwidth as well as the number of features required are tested.

**Summary Of The Review:**

The pedagogical aspect of this paper is certainly an appreciable contribution. This paper also contributes to understanding how certain biases of DNNs can be dealt with in deep RL, but only within a limited scope. While it certainly refines our intuition and could help the field focus its research in the right areas, the novelty of the concepts introduced in this paper is fairly limited.

I think that, as it is, this paper meets the bar for acceptance, but its potential seems much higher.

---

> ### Author Response · Authors · 2021-11-23
> **Author's reply (1/2)**
>
> The authors would like to thank reviewer Tezl for their kind review.
>
> ## On The Main Contribution of This Paper
>
> We are motivated by the key issue in deep reinforcement learning -- **learning to approximate the value with a neural network**. This is **a fundamental issue in RL that takes many different forms, and remains open even after years of work from a number of high-impact publications** [^hesselt] [^fujimoto]. The majority of these prior work focus on the learning procedure. Whereas we set out to show via our motivating example, that the problem primarily lies within the function approximator itself. This is our first contribution.
>
> [^hesselt]: Hasselt, H. (2010) ‘Double Q-learning’, in Lafferty, J. et al. (eds) Advances in Neural Information Processing Systems. Curran Associates, Inc.
> [^fujimoto]: Fujimoto, S., van Hoof, H. and Meger, D. (2018) ‘Addressing Function Approximation Error in Actor-Critic Methods’, arXiv [cs.AI].
> [^rahaman]: Rahaman, N. et al. (2018) ‘On the Spectral Bias of Neural Networks’, arXiv [stat.ML]. Available at: http://arxiv.org/abs/1806.08734.
>
> Kernel regression, and neural tangent kernel in particular are still fairly new to the general deep learning community. To the best of our knowledge, at the time of submission to open review, we are the **first paper to introduce such concepts to reinforcement learning**. If recent publications on the simplicity bias is of any indication, we believe this is just the first in a series of work in this direction. This is why we would like to release our research codebase as soon as our paper is accepted to maximize our impact. Our codebase covers additional comparisons between all kernels mentioned by reviewers (see next section), and a setup to evaluat over all of the MuJoCo and DeepMind control domains, using TD3, SAC, and DDPG.
>
> We would also like to note that our spectral analysis treatment to the fitted Q iteration procedure is novel. Making this connection between learning theory [^rahaman], and the parculiarities of value approximation through the *spectral lense* is our **second contribution**. In the end, the specific parameterization that we used to produce the large gain is just one of many that could improve the spectral signature of the network. Therefore regarding your comment:
>
> > This paper proposes a novel parameterization for MLPs, Random Fourier Features (RFF), which amount to using a sinusoidal activation, and initializing parameters to capture different frequencies
>
> The authors definitely appreciate your kind works, but would like to emphasize that our analytical contribution to illustrating and formulating the problem, and making the connection, together with our strong empirical gain are the key results.
>
> ## Additional Results On "Large Scale" Problems
>
> > The empirical evaluation on "large scale" problems is limited
>
> **we have included new experimental results with the state-of-the-art TD3 learning algorithm** in section 2 of the appendix. We report results on the DMC Hopper domain in this updated draft, but are in the process of scaling up to common DMC domains.
>
> In otherwords, we extend our paper along all three axes below:
> 1. Include more large scale problems such as DeepMind control suite domains. We just finished Hopper, and are in the process of including more.
> 2. Include all of the kernels that have been mentioned: Tanick, Siren, and learned fourier features.
> 3. Include more actor-critic baselines. We include results on TD3, the state-of-the-art algorithm, and show a significant gain and a new state-of-the-art.
>
> We are in the process of finishing up these large number of experiments. The updated draft contains a limited set of these new results.
>
> ## Comparing Between Learned Vs Fixed Random Fourier Features
>
> In appendix 2, we include additional comparison between the following kernels:
> 1. vanilla MLP
> 2. Random Fourier Features (ours) that uses sine activation with random phase-shift
> 3. RFF (Tanick) that uses sine and cosine pairs
> 4. LFF (learned fourier features) (ours)
> 5. Siren network that uses stacked learned Fourier layers and Sitzmann initialization
>
> We offer these results on Ant, Cheetah, and Humaoid domains. Sirens in particular works poorly, which agrees with the observations that the network performance is quite sensitive to the weight initialization.

---

> > ### Author Response · Authors · 2021-11-23
> > **Author's reply (2/2)**
> >
> > ## New Results on The Effective Rank
> >
> > We were looking for a quantitative way to show that random Fourier features improve the complexity of the learned value function. We do so in a new experiment by computing the effective rank [^roy_2007] of the penmultimate layer of the value function over data in the replay buffer. Our result (section A.3 in the appendix) indicate value functions equipped with the random fourier features indeed learn a higher rank solution, beside obtaining superior perfomance.
> >
> > | Q function rank | RFAC            | DDPG            |
> > | --------------- | --------------- | --------------- |
> > | Ant             | $2.86 \pm 1.32$ | $1.08\pm 1.07$  |
> > | Humanoid        | $2.94 \pm 1.26$ | $1.49 \pm 1.22$ |
> > | Half Cheetah    | $2.94 \pm 1.12$ | $1.01 \pm 1.00$ |
> >
> >
> >
> > ## Other Comments
> >
> > We have also fixed all of the citation entries with the correct conference proceedings.
> >
> > [^roy_2007]: Roy, O. and Vetterli, M. (2007) ‘The effective rank: A measure of effective dimensionality’, in 2007 15th European Signal Processing Conference, pp. 606–610.

---

> ### Author Response · Authors · 2021-11-29
> **Looking forward to further discussions!**
>
> Dear Reviewer,
>
> Thank you for your time and effort in reviewing our work. We have provided detailed clarification and additional experiments to address the issues raised in your comments. If our response has addressed your concerns, we would be grateful if you could re-evaluate our work.
>
> If you have any additional questions or comments, we would be happy to have further discussions.
>
> Thanks,
>
> The authors

---

> > ### Comment · Reviewer_TezL · 2021-11-29
> > **Response**
> >
> > Thanks for all the responses, it will help the decision discussion.
> >
> > I do appreciate that limited novelty doesn't mean that no contribution is being made, and there are interesting new links made in this paper with other aspects of the credit assignment problem. The new results will also help position this paper.

---

### Official Review · Reviewer_cq8o · 2021-11-01

**Correctness:** 3
**Technical Novelty And Significance:** 2
**Empirical Novelty And Significance:** 2
**Recommendation:** 3
**Confidence:** 4

**Main Review:**

Discussion period update: After reading the other reviews (note: no author response), my overall score has not changed.

-This work has limited novelty. The core architecture is basically pulled from existing work. One difference is a change in weight initialization, but the initialization schemes are not directly compared within the experiments. Additionally, if I understand correctly, both schemes still require a hyperparameter (with appropriate choice of hyperparameter, the schemes are the same). The other difference is that this work focuses on DRL.

-The baselines include a decent set of non-Fourier baselines. However, this work does not compare to architectures from Sitzmann 2020 and Tancik 2020 (though these works are cited). As a minor comment, including "dense+deep" and "wide" would be better, as would including more discussion on the number of parameters vs depth of different baselines.

-This work includes claims about "evading the spectral bias" and, in some sections, uses performance to "show" the efficacy of the proposed method. However, improved performance does not necessarily mean the bias issue has been reduced.

Other comments:

-The MLP example (paragraph 2) in the introduction is presented in an unclear way. Insufficient information is provided to show that the training process was not the reason for poor performance.

-The third paragraph of the introduction is better placed in a later section.

-For Figure 2 and Figure 7, showing the learned function does not convey the difference between the true and learned functions. Showing the absolute difference between the two would be more informative.

-Much of Section 2 ("Preliminaries") is not relevant to this work's contributions. This section should be substantially condensed.

-Reorganizing Section 3.1 would improve clarity. Additionally, the first paragraph largely repeats content from earlier in the paper.

-The Mountain Car experiment is better placed elsewhere in the text.

-If I understand correctly, when varying bandwidth, the number of parameters in the network also changes. This should be considered when performing the ablation studies. Additionally, the SIREN-style selection of width should be explicitly compared to the proposed method.

-Related Work is best placed earlier in the paper.

This work would benefit from another editing pass:

-"Given that a more gradient updates" -> "Given that more gradient updates"

-"as pointed in Fu et al." -> "as pointed out in Fu et al."

-"is called the spectral norm" -> "is the spectral norm"

-"in figure 4" -> "in Figure 4"

-"Xiavier-like" -> "Xavier-like"

-"deepr reinforcement learning" -> "deep reinforcement learning"

-"In figure 7e" -> "In Figure 7"

-"its performance vary a lot" -> "its performance varies a lot"

**Summary Of The Paper:**

This work proposes Random Fourier Networks (RFNs), MLPs with random Fourier featurization within the first layer. RFNs are supposed to learn high-frequency components of the value function more quickly than standard MLPs. The authors present an initialization scheme for the Fourier layer that is better-suited for high-dimensional inputs. RFNs are compared to non-Fourier-feature baselines on high-dimensional environments.

**Summary Of The Review:**

This work overstates its novelty. Using sinusoidal activations on RL tasks is of limited novelty, and no comparisons are made to related methods / few changes are made that are specific to RL.

---

> ### Author Response · Authors · 2021-11-23
> **Author's reply (1/3)**
>
> Thanks for your feedback and comments. We have taken another writing pass and improved upon the grammatical/clarity issues highlighted by the reviewer. We now address specific points below:
>
> ## Varying Bandwidth Does Not Affect Number of Parameters
>
> > If I understand correctly, when varying bandwidth, the number of parameters in the network also changes. This should be considered when performing the ablation studies.
>
> The authors regret the lack of clarity, and would like to clarify: The bandwidth parameter **does not** affect the total number of parameters. Instead, for an input dimension N, and a fixed feature dimension M, the weight matrix for the random Fourier Feature layer is *identical* to those of a single linear layer. Both the weight matrix $W_{[N, M]}$, and the bias $c_{[M]}$ are the same as a typical linear layer.
>
> We have updated the corresponding section in the paper and reproduced some of the details below on parameterization:
>
> The random Fourier feature layer is parameterized as
> $$
> f(x) = \sin ( W @ x + c )
> $$
> where the random matrix $W$ is sampled from a normal distribution $\mathcal N(0, 2\pi b/n)$. We refer to the parameter $b$ as the bandwidth and normalize it with the input dimension $n$. We sample the biases $c$ from a uniform distribution $\mathcal{U}(-\pi,\pi)$.
>
> Hence the bandwidth parameter does not affect the number of the parameters.
>
> > Additionally, the SIREN-style selection of width should be explicitly compared to the proposed method.
>
> The difference between these initialization schemes is in the input dimension. Since we won't get to change the input dimension of the problem, we demonstrate the benefit of our initialization scheme by showing that a small range of $b$ works consistently well across a variety of RL domains that differ in the dimensionality of the observation space by two magnitudes. We only had to tune $b$ between 5 and 15 during our experiments, whereas with the SIREN initialization, this range will be magnitudes bigger.
>
> It is also important to note that Siren is known to be sensitive to initialization parameters in practice. We now offer SIREN as one of the kernels that we compare in Appendix 3.

---

> > ### Author Response · Authors · 2021-11-23
> > **Author's reply (2/3)**
> >
> > ## On Novelty
> >
> > Value approximation using neural networks is at the center of a number of long-standing problems in deep reinforcement learning. It is true that we are inspired by Tancik et al. and Sitzmann et al. In fact, as we mentioned in the text, using fourier features in RL is an old idea [^1] that dates all the way back to 2011. A related line of work, proto-value functions generalizes Fourier bases to high-dimension and non-euclidean manifolds [^2], which dates back to 2007. The existence of these prior works however, do not negate our novelty and contribution.
> >
> > Our submission makes a novel connectiong between underfitting and value over-estimation in RL [^fujimoto_2018] with the RFF and SIREN kernels. The neural tangent kernel tooling itself is quite new [^NTK], and it wasn't until last year when the graphics community first introduced NTK as a way to understand the importance of RFF in NeRF [^tanick_2020].
> >
> >
> > [^1]: Konidaris, G., Osentoski, S. and Thomas, P. (2011) ‘Value Function Approximation in Reinforcement Learning Using the Fourier Basis’, *Proceedings of the AAAI Conference on Artificial Intelligence*, 25(1), pp. 380–385.
> >
> > [^2]: Mahadevan, S. (2007) *Proto-value functions: A Laplacian framework for learning representation and control in Markov decision processes*.
> >
> > [^NTK]: Jacot, A., Gabriel, F. and Hongler, C. (2018) ‘Neural Tangent Kernel: Convergence and Generalization in Neural Networks’, arXiv [cs.LG]. Available at: http://arxiv.org/abs/1806.07572.
> >
> > [^tanick_2020]: Tancik, M., Srinivasan, P. P. and Mildenhall, B. (2020) ‘Fourier features let networks learn high frequency functions in low dimensional domains’
> >
> > [^fujimoto_2018]: Fujimoto, S., van Hoof, H. and Meger, D. (2018) ‘Addressing Function Approximation Error in Actor-Critic Methods’, arXiv [cs.AI].
> >
> > > *This work has limited novelty. The core architecture is basically pulled from existing work. One difference is a change in weight initialization, but the initialization schemes are not directly compared within the experiments. Additionally, if I understand correctly, both schemes still require a hyperparameter (with an appropriate choice of hyperparameter, the schemes are the same). The other difference is that this work focuses on DRL.*
> >
> > Indeed our idea is quite simple, but it has a rich theory behind it and works substantially better than prior methods on more complex domains.
> >
> > **To summarize Our Contribution**, our submission
> >
> > 1. Introduce a spectral analysis angle to understanding neural approximation
> > 2. Identify underfitting in Q-learning is not occurring because of batch TD-learning or stochastic approximations, but in fact also occurs when we learn Q-function using supervised learning
> > 3. We have compared against prior (arguably more complex) approaches addressing this problem, such as D2RL [^D2RL]
> >
> > This is a story that has repeated multiple times in Deep Learning. The prime examples is the AlexNet paper — the ideas in the paper were not novel — it was just that scaling it up to large datasets outperformed existing methods by a large margin. One could have argued that AlexNet is a scaled up version of LeNet and therefore not novel and the several data normalization / data augmentation techniques exist prior work. We are not claiming our paper is even close to being as revolutionary as AlexNet, but just highlighting that methods that leverage prior work, but are empirically shown to solve important problems should be credited instead of holding simplicity / technical novelty as a barrier to publication.
> >
> > [^D2RL]: Sinha, S. et al. (2020) ‘D2RL: Deep Dense Architectures in Reinforcement Learning’, arXiv [cs.LG]. Available at: http://arxiv.org/abs/2010.09163.

---

> > > ### Author Response · Authors · 2021-11-23
> > > **Author's reply (3/3)**
> > >
> > > ## Including Additional Kernels and Test Environments
> > >
> > > > *The baselines include a decent set of non-Fourier baselines. However, this work does not compare to architectures from Sitzmann 2020 and Tancik 2020 (though these works are cited). As a minor comment, including "dense+deep" and "wide" would be better, as would including more discussion on the number of parameters vs depth of different baselines*
> > >
> > > The authors greatly appreciate your feedback, and would like to offer a comprehensive set of empirical studies over standard domains and kernels that have appeared in the literature. To this end, we have expanded the experiments in the updated draft along the following axes:
> > > 1. **Domain-wise, we now include the DMC domain** Hopper in Appendix 3. More DMC domains are currently running.
> > > 2. **Kernel-wise**, we now offer a comparison between RFF from Tanick, Siren, Learned Fourier Features (LFF) besides the vanilla MLP and D2RL network. We now include this in Appendix 2.
> > > 3. **Base algorithm-wise**, we applied our method to TD3! We include the comparison between RFF and MLP with TD3 on Ant, and show that there remains a large gain with the random fourier features.
> > >
> > > ## Measuring the Spectral Bias via the Effective Rank
> > >
> > > > This work includes claims about "evading the spectral bias" and, in some sections, uses performance to "show" the efficacy of the proposed method. However, improved performance does not necessarily mean the bias issue has been reduced.
> > >
> > > To address your concern, we use the *effective rank* [^roy_2007], which is a standard technique that is used in [^huh] and [^kumar] to measure the complexity of a learned neural network. We offer the definition of the effective rank in Section 2.1 (now 3.1 in the updated draft).
> > >
> > > Our experiment show that random Fourier features significantly improves the effective rank of the value function during learning.
> > >
> > >
> > > | Q function rank | RFAC            | DDPG            |
> > > | --------------- | --------------- | --------------- |
> > > | Ant             | $2.86 \pm 1.32$ | $1.08\pm 1.07$  |
> > > | Humanoid        | $2.94 \pm 1.26$ | $1.49 \pm 1.22$ |
> > > | Half Cheetah    | $2.94 \pm 1.12$ | $1.01 \pm 1.00$ |
> > >
> > >
> > > [^roy_2007]: Roy, O. and Vetterli, M. (2007) ‘The effective rank: A measure of effective dimensionality’, in 2007 15th European Signal Processing Conference, pp. 606–610.
> > >
> > > [^huh]: Huh, M. et al. (2021) ‘The Low-Rank Simplicity Bias in Deep Networks’.
> > >
> > > [^kumar]: Kumar, A. et al. (2020) ‘Implicit under-parameterization inhibits data-efficient deep reinforcement learning’, arXiv preprint arXiv:2010.14498.
> > >
> > > ## Additional Details on the Motivating Example
> > >
> > > We have expanded the appendix (Appendix 6) to include details on this experiment. We include those details here for completenes.
> > >
> > > **Offline data generation** We divided the 1 dimensional state space into 1000 discrete bins and used the midpoints of the bins as initial states. We then took both the actions from these initial states to get corresponding next states. We used the dataset of these transitions (2000 total transitions) as our offline dataset.
> > >
> > > **Optimization details** We use 4-layer MLP with ReLU Activation, with 400 latent neurons. We use Adam optimization with a learning rate of 1e-4, and optimize for 400 epochs. We use gradient descent with a batch size of 200. For a deeper network, we use a 12-layer MLP, with 400 latent neurons but keep other optimization hyperparameters fixed. To show that longer training period help MLPs evade the spectral bias, we use a 4-layer MLP, with 400 latent neurons and train it for 2000 epochs. All the optimization hyperparameters are kept the same.
> > >
> > >
> > > ## Additional Comments
> > > > For Figure 2 and Figure 7, showing the learned function does not convey the difference between the true and learned functions. Showing the absolute difference between the two would be more informative.
> > >
> > > If the value function is a single curve, it would have been easy to plot the learned value function on top of the ground-truth value function. However in this case the q-function is two curves instead of one. The authors experimented with plotting all four curves in a single plot (two fitted, two ground-truth) but it was visually difficult to parse.
> > >
> > > The true value function for those learned curves in Figure 2 can be found in Figure 1 (along with the dynamics). The true value function in Figure 7 is sub-figure 7(e). We use the function produced by tabular value iteration as the ground truth in this toy domain.
> > >
> > > > The Mountain Car experiment is better placed elsewhere in the text.
> > >
> > > We have re-arranged the Mountain Car experiment, to place it in the results section (section 4). We have also updated the overall structure in response to comments from reviewer fVzS.

---

> ### Author Response · Authors · 2021-11-29
> **Looking forward to further discussions!**
>
> Dear Reviewer cq8o,
>
> Thank you for your time and effort in reviewing our work. We have provided detailed clarification and additional experiments to address the issues raised in your comments. If our response has addressed your concerns, we would be grateful if you could re-evaluate our work.
>
> If you have any additional questions or comments, we would be happy to have further discussions.
>
> Thanks,
>
> The authors

---

### Author Response · Authors · 2021-11-23
**Common reply to reviewers and the area chair (1/2)**

To all reviewers cq80, Tezl, fVzS and the area chair:

## On Novelty

Value approximation using neural networks is at the center of a number of long-standing problems in deep reinforcement learning. It is true that we are inspired by Tancik et al. and Sitzmann et al. In fact, as we mentioned in the text, using fourier features in RL is an old idea [^1] that dates all the way back to 2011. A related line of work, proto-value functions generalizes Fourier bases to high-dimension and non-euclidean manifolds [^2], which dates back to 2007. The existence of these prior works however, do not negate our novelty and contribution.

On the contrary our submission makes a novel connection between underfitting and value over-estimation in RL [^fujimoto_2018] with the RFF and SIREN kernels. The neural tangent kernel tooling itself is quite new [^NTK], and it wasn't until last year when the graphics community first introduced NTK as a way to understand the importance of RFF in NeRF [^tanick_2020].

**To Summarize Our Contribution**, our submission

1. Introduce a spectral analysis angle to understanding neural value approximation
2. Identify underfitting in Q-learning is not occurring because of batch TD-learning or stochastic approximations, but in fact also occurs when we learn Q-function using supervised learning
3. We have compared against prior (arguably more complex) approaches addressing this problem, such as D2RL [^D2RL], and show significant empirical gain.
4. We produce state-of-the-art performance on challenging tasks such as Ant and Humanoid

Our work tackles an import issue in deep reinforcement learning. We demonstrate through our experiments that this simple change scales well to much more complex, high-dimensional domains.

This is a story that has repeated multiple times in Deep Learning. The prime examples is the AlexNet paper — the ideas in the paper were not novel — it was just that scaling it up to large datasets outperformed existing methods by a large margin. One could have argued that AlexNet is a scaled up version of LeNet and therefore not novel and the several data normalization / data augmentation techniques exist prior work. We are not claiming our paper is even close to being as revolutionary as AlexNet, but just highlighting that methods that leverage prior work, but are empirically shown to solve important problems should be credited instead of holding simplicity / technical novelty as a barrier to publication.



[^1]: Konidaris, G., Osentoski, S. and Thomas, P. (2011) ‘Value Function Approximation in Reinforcement Learning Using the Fourier Basis’, *Proceedings of the AAAI Conference on Artificial Intelligence*, 25(1), pp. 380–385.

[^2]: Mahadevan, S. (2007) *Proto-value functions: A Laplacian framework for learning representation and control in Markov decision processes*.

[^NTK]: Jacot, A., Gabriel, F. and Hongler, C. (2018) ‘Neural Tangent Kernel: Convergence and Generalization in Neural Networks’, arXiv [cs.LG]. Available at: http://arxiv.org/abs/1806.07572.

[^tanick_2020]: Tancik, M., Srinivasan, P. P. and Mildenhall, B. (2020) ‘Fourier features let networks learn high frequency functions in low dimensional domains’

[^fujimoto_2018]: Fujimoto, S., van Hoof, H. and Meger, D. (2018) ‘Addressing Function Approximation Error in Actor-Critic Methods’, arXiv [cs.AI].

[^D2RL]: Sinha, S. et al. (2020) ‘D2RL: Deep Dense Architectures in Reinforcement Learning’, arXiv [cs.LG]. Available at: http://arxiv.org/abs/2010.09163.


------


## Including Additional Kernels and Test Environments

> The baselines include a decent set of non-Fourier baselines. However, this work does not compare to architectures from Sitzmann 2020 and Tancik 2020 (though these works are cited).

The authors greatly appreciate your feedback, and would like to offer a comprehensive set of empirical studies over standard domains and kernels that have appeared in the literature. To this end, we have expanded the experiments in the updated draft along the following axes:
1. **Domain-wise, we now include the DMC domain** Hopper in Appendix 3. More DMC domains are currently running.
2. **Kernel-wise**, we now offer a comparison between RFF from Tanick, Siren, Learned Fourier Features (LFF) besides the vanilla MLP and D2RL network. We now include this in Appendix 2.
3. **Base algorithm-wise**, we applied our method to TD3! We include the comparison between RFF and MLP with TD3 on Ant, and show that there remains a large gain with the random fourier features.

---

> ### Author Response · Authors · 2021-11-23
> **Common reply to reviewers and the area chair (2/2)**
>
> ## Measuring the Spectral Bias via the Effective Rank
>
> > This work includes claims about "evading the spectral bias" and, in some sections, uses performance to "show" the efficacy of the proposed method. However, improved performance does not necessarily mean the bias issue has been reduced.
>
> To address your concern, we use the *effective rank* [^roy_2007], which is a standard technique that is used in [^huh] and [^kumar] to measure the complexity of a learned neural network. We offer the definition of the effective rank in Section 2.1 (now 3.1 in the updated draft).
>
> Our experiment show that random Fourier features significantly improves the effective rank of the value function during learning.
>
>
> | Q function rank | RFAC            | DDPG            |
> | --------------- | --------------- | --------------- |
> | Ant             | $2.86 \pm 1.32$ | $1.08\pm 1.07$  |
> | Humanoid        | $2.94 \pm 1.26$ | $1.49 \pm 1.22$ |
> | Half Cheetah    | $2.94 \pm 1.12$ | $1.01 \pm 1.00$ |
>
>
> [^roy_2007]: Roy, O. and Vetterli, M. (2007) ‘The effective rank: A measure of effective dimensionality’, in 2007 15th European Signal Processing Conference, pp. 606–610.
>
> [^huh]: Huh, M. et al. (2021) ‘The Low-Rank Simplicity Bias in Deep Networks’.
>
> [^kumar]: Kumar, A. et al. (2020) ‘Implicit under-parameterization inhibits data-efficient deep reinforcement learning’, arXiv preprint arXiv:2010.14498.

---

### Author Response · Authors · 2021-11-28
**Common reply to reviewers: New results on DeepMind Control domains**

# New, Comprehensive Comparison on EIGHT DeepMind Control Suite Tasks

The authors would like to bring your attention to our results on **eight** DeepMind control suite tasks. This makes our experimental evaluation much more thorough and comprehensive. We reproduce the performance in table form below, where we order the domains by their state-action space complexity. An overall trend is that more complex tasks with higher input dimensions benefit more from random Fourier features.

To report statistical significance, we use Welch's t-test per the recommendation by [^colas_2019]. $p\lt 0.05$ is considered significant.

## Fourier Features on DDPG and DeepMind Control Tasks

| Name             | Observation   | Action   | DDPG + MLP        | DDPG + RFF                    | p_value  |
| ---------------- | ------------- | -------- | ----------------- | ----------------------------- |----------|
| Acrobot-swingup  | Box(6,)       | Box(1,)  | $174.3 \pm 92.6$  | $172.9 \pm 108.2$             | 0.98     |
| Finger-turn_hard | Box(12,)      | Box(2,)  | $816.8 \pm 92.1$  | $864.1 \pm 53.8$              | 0.28     |
| Hopper-hop       | Box(15,)      | Box(4,)  | $282.1 \pm 189.9$ | $353.1 \pm 106.9$             | 0.49     |
| Cheetah-run      | Box(17,)      | Box(6,)  | $787.2 \pm 77.9$  | $\boldsymbol{890.5 \pm 18.9}$ | 0.0076   |
| Walker-run       | Box(24,)      | Box(6,)  | $750.9 \pm 17.9$  | $\boldsymbol{772.1 \pm 11.5}$ | 0.04     |
| Humanoid-run     | Box(67,)      | Box(21,) | $1.3 \pm 0.1$     | $\boldsymbol{177.9 \pm 14.7}$ | 1.11e-5  |
| Quadruped-run    | Box(78,)      | Box(12,) | $290.9 \pm 119.3$ | $\boldsymbol{799.4 \pm 75.9}$ | 5.78e-5  |
| Quadruped-walk   | Box(78,)      | Box(12,) | $311.1 \pm 143.4$ | $\boldsymbol{939.6 \pm 12.3}$ | 0.00039  |

## Fourier Features on SAC and DeepMind Control Tasks

| Name             | Observation   | Action   | SAC + MLP                     | SAC + RFF                     | p_value |
| ---------------- | ------------- | -------- | ----------------------------- | ----------------------------- | ------- |
| Acrobot-swingup  | Box(6,)       | Box(1,)  | $101.7 \pm 87.2$              | $\boldsymbol{219.8 \pm 61.4}$ | 0.03    |
| Finger-turn_hard | Box(12,)      | Box(2,)  | $\boldsymbol{836.2 \pm 80.8}$ | $552.2 \pm 182.0$             | 0.015   |
| Hopper-hop       | Box(15,)      | Box(4,)  | $147.8 \pm 88.9$              | $\boldsymbol{322.8 \pm 27.8}$ | 0.0087  |
| Cheetah-run      | Box(17,)      | Box(6,)  | $829.3 \pm 59.8$              | $\boldsymbol{881.0 \pm 12.9}$  | 0.01    |
| Walker-run       | Box(24,)      | Box(6,)  | $\boldsymbol{852.4 \pm 7.1}$  | $811.7 \pm 22.1$              | 0.012   |
| Humanoid-run     | Box(67,)      | Box(21,) | $191.5 \pm 40.3$              | $\boldsymbol{261.6 \pm 35.4}$ | 0.022   |
| Quadruped-run    | Box(78,)      | Box(12,) | $582.2 \pm 216.9$             | $\boldsymbol{919.4 \pm 20.4}$ | 0.023   |
| Quadruped-walk   | Box(78,)      | Box(12,) | $485.1 \pm 280.5$             | $\boldsymbol{944.6 \pm 14.1}$ | 0.02    |

## Images of the Learning Curves

We attach updated learning curves below:
- DDPG on DeepMind Control: https://imgur.com/a/OhcwI7A
- SAC on DeepMind Control: https://imgur.com/a/yivWdTj

[^colas_2019]: Colas, C., Sigaud, O. and Oudeyer, P.-Y. (2019) ‘A Hitchhiker’s Guide to Statistical Comparisons of Reinforcement Learning Algorithms’. Available at: https://www.semanticscholar.org/paper/d36a59a4d6769e4ec8e71bf17c28434017043b63 (Accessed: 26 November 2021).

---

### Decision · Program_Chairs · 2022-01-20

**Decision:**

Accept (Poster)

**Comment:**

The paper points out an interesting and, to me unexpected, problem when learning Q-functions to do with spectral bias. Figures 1 and 2 are quite striking. The diagnosis and proposed solution elegantly combines ideas from NTKs and NeRFs. The proposed random Fourier actor-critic performs well in practice. The main problem reviewers had in the end is that the authors added substantial new empirical results too late to review thoroughly.